# ADAM17-triggered TNF signalling protects the ageing *Drosophila* retina from lipid droplet-mediated degeneration

Sonia Muliyil[†] (ID), Clémence Levet, Stefan Düsterhöft[††] (ID), Iqbal Dulloo, Sally A Cowley (ID) & Matthew Freeman[*] (ID)

## Abstract

**Animals have evolved multiple mechanisms to protect themselves from the cumulative effects of age-related cellular damage. Here, we reveal an unexpected link between the TNF (tumour necrosis factor) inflammatory pathway, triggered by the metalloprotease ADAM17/TACE, and a lipid droplet (LD)-mediated mechanism of protecting retinal cells from age-related degeneration. Loss of ADAM17, TNF and the TNF receptor Grindelwald in pigmented glial cells of the *Drosophila* retina leads to age-related degeneration of both glia and neurons, preceded by an abnormal accumulation of glial LDs. We show that the glial LDs initially buffer the cells against damage caused by glial and neuronally generated reactive oxygen species (ROS), but that in later life the LDs dissipate, leading to the release of toxic peroxidated lipids. Finally, we demonstrate the existence of a conserved pathway in human iPS-derived microglia-like cells, which are central players in neurodegeneration. Overall, we have discovered a pathway mediated by TNF signalling acting not as a trigger of inflammation, but as a cytoprotective factor in the retina.**

**Keywords** ADAM17; Glia; lipid droplet; neurodegeneration; reactive oxygen species
**Subject Categories** Neuroscience; Signal Transduction
**The EMBO Journal (2020) 39: e104415**

## Introduction

Diseases of ageing are often caused by stress-induced cellular degeneration that accumulates over time (Campisi, 2013; Lopez-Otin *et al*, 2013). The intrinsic causes of such damage are widespread but include toxic build-up of misfolded and aggregated proteins, as well as oxidative stress caused by the production of reactive oxygen species (ROS), which are by-products of metabolic and other physiological activity (Squier, 2001; Davalli *et al*, 2016; Klaips *et al*, 2018). Cells have evolved multiple processes to protect themselves against these potentially damaging stresses, including well-characterised systems like the unfolded protein response, endoplasmic reticulum (ER)-associated degradation, and reactive oxygen species (ROS) scavenging enzymes (Wellen & Thompson, 2010; Walter & Ron, 2011; Bravo *et al*, 2013). Recently, lipid droplets (LDs) have begun to be implicated in the machinery of stress protection (Bailey *et al*, 2015; Van Den Brink *et al*, 2018). The significance of this role of LDs has been most studied in the fruit fly *Drosophila*. In central nervous system stem cell niches, elevated ROS levels induce the formation of LDs, which appear to sequester polyunsaturated acyl chains, protecting them from the oxidative chain reactions that generate toxic peroxidated species (Bailey *et al*, 2015). In another context, however, LDs are part of the cellular damage causing pathway: ROS generated by defective mitochondria in the *Drosophila* retinal neurons induces the formation of LDs in adjacent glial cells, and these LDs later contribute to glial and neuronal degeneration (Liu *et al*, 2015). More recent work has illustrated that toxic fatty acids produced by activated neurons in culture can be taken up by astrocytes via endocytosis, where they get subsequently catabolised by mitochondrial beta-oxidation (Ioannou *et al*, 2019). Although the overall significance of these mechanisms, and how they are integrated, remains to be understood, it is clear that the long-held view of LDs as primarily inert storage organelles is no longer tenable (Olzmann & Carvalho, 2019). Furthermore, beyond their role in regulating cell survival and death, LDs are increasingly found to act in other cellular processes, including acting as platforms for the assembly of viruses, and modulators of cell signalling (Boulant *et al*, 2008; Cheung *et al*, 2010; Li *et al*, 2014; Sandoval *et al*, 2014; Welte & Gould, 2017). They also form intimate contacts with ER and mitochondria (Schuldiner & Bohnert, 2017; Thiam & Dugail, 2019) and act inside the nucleus (Layerenza *et al*, 2013; Uzbekov & Roingeard, 2013; Soltysik *et al*, 2019).

In addition to the cellular mechanisms of protection against stresses, there is also protection at the level of the whole organism. This higher order, coordinated protection is primarily mediated by

Sir William Dunn School of Pathology, University of Oxford, Oxford, UK
*Corresponding author. Tel: +44 01993 704986; E-mail: matthew.freeman@path.ox.ac.uk
†Present address: Elsevier, Oxford, UK
††Present address: Institute of Pharmacology and Toxicology, Medical Faculty, RWTH Aachen University, Aachen, Germany

the inflammatory and immune systems (Chen *et al*, 2018; Ferrucci & Fabbri, 2018; Franceschi *et al*, 2018; O'Neil *et al*, 2018). Inflammatory signalling pathways are increasingly understood to have relevance to an extraordinary range of biology, extending far beyond classical inflammation, and including age-related damage. This is highlighted by the ever-growing list of diseases associated with inflammation including, for example, neurodegeneration, multiple sclerosis and other neurological diseases; metabolic pathologies like type 2 diabetes and obesity/metabolic syndrome; and atherosclerosis (Ferrucci & Fabbri, 2018). The signalling molecules associated with these myriad functions are equally diverse, but TNF (tumour necrosis factor) stands out as being a primary trigger of much classical and non-classical inflammatory signalling (Sedger & McDermott, 2014). Like many cytokines and growth factors, TNF is synthesised with a transmembrane (TM) anchor, and its release as an active signal is triggered by the proteolytic cleavage of the extracellular domain from its TM anchor by the "shedding" protease ADAM17 (a disintegrin and metalloproteinase 17; also known as TACE, TNF alpha converting enzyme) (Black *et al*, 1997). Its function of being the essential trigger of all TNF signalling makes ADAM17 a highly significant enzyme. But in fact its importance is even greater because, in addition to shedding TNF, it is also responsible for the release of many other signalling molecules including EGF (epidermal growth factor) receptor ligands (Sahin *et al*, 2004; Baumgart *et al*, 2010; Dang *et al*, 2013). Consistent with this central role in inflammation and a wide range of other cellular events, ADAM17 has been extensively studied, both from a fundamental biological perspective and also as a drug target.

The fruit fly *Drosophila*, an important model organism for revealing the molecular, cellular and genetic basis of development, is increasingly used to investigate conserved aspects of human physiology and even disease mechanisms. With this motivation, we have investigated in flies the pathophysiological role of ADAM17. We report the first ADAM17 mutation in *Drosophila* and find that the mutant flies exhibit abnormally high lipid droplet accumulation followed by severe age- and activity-dependent neurodegeneration in the adult retina. These observations have uncovered a new cytoprotective pathway mediated by soluble TNF acting not as a trigger of inflammation, but as a trophic survival factor for retinal glial cells. We have also shown that inhibition of ADAM17 in human iPSC-derived microglia-like cell lines leads to the same cellular phenomena as seen in fly retinas—lipid droplet accumulation, ROS production and generation of peroxidated lipids—suggesting that ADAM17 may have an evolutionarily conserved cytoprotective -function.

# Results

## Loss of ADAM17 in glial cells triggers age-dependent degeneration

To investigate the full range of its pathophysiological functions, we made *Drosophila* null mutants of ADAM17 using CRISPR/Cas9. The knockout flies did not display any obvious defects during development and were viable as adults. They did, however, have reduced lifespan, indicating potential physiological defects. We pursued this possibility by more detailed analysis of the adult ADAM17 mutant ($ADAM17^{-/-}$) retina, a tissue widely used for studying age-related degeneration of the nervous system (Morante & Desplan, 2005). The retina of the *Drosophila* compound eye comprises ommatidial units of eight photoreceptors, each containing a light-collecting organelle called the rhabdomere, that project axons into the brain. Photoreceptor cell bodies are surrounded by pigmented glial cells (PGCs) believed to provide metabolic support to the neurons (Edwards & Meinertzhagen, 2010; Liu *et al*, 2015). $ADAM17^{-/-}$ retinas exhibited extensive degeneration of both photoreceptor neurons and the surrounding PCGs in 5-week-old flies (Fig 1A–C and L). The same result was seen with a heterozygous mutant allele in combination with a chromosome deficiency that includes the ADAM17 locus ($ADAM17^{-}$/Df) (Fig 1D and L). This supports the conclusion that the degeneration was indeed caused by the loss of ADAM17. To further confirm the role of ADAM17 in the degeneration phenotype, we analysed the siRNA knockdown of ADAM17 using a retinal specific driver, *GMR-GAL4*. This too showed glial and neuronal degeneration (Fig 1E, F and L).

An ADAM17-specific antibody revealed that expression of the enzyme is strongly enriched in the PGCs that surround the photoreceptor neurons (Fig 1M and N). This expression pattern hinted that ADAM17 function may be required specifically in glial cells, so we tested this idea by cell-type specific knockdown. ADAM17 siRNA, expressed specifically in neurons with *elav-GAL4*, produced no phenotype (Fig 1G, H and L, and EV1L). In sharp contrast, glial-specific knockdown with a sparkling-GAL4 driver caused severe and characteristic degeneration (Fig 1I, J and L, and EV1L). We confirmed that ADAM17 is not needed in neurons with another neuron-specific GAL4 driver, Rh1-GAL4, which drives expression specifically in photoreceptors (Fig EV1A, B, K and L). The specific requirement of ADAM17 in glia was further demonstrated by showing that glial expression of ADAM17 was sufficient to rescue the degeneration phenotype of the mutant (Fig EV1C, D and K). We also investigated the specificity of the phenotype by examining fly mutants of the closely related ADAM metalloprotease, Kuzbanian,

**Figure 1. Glia-specific loss of ADAM17 results in age-associated retinal degeneration.**

A     Diagram of a tangential (left) and horizontal (right) section of the adult *Drosophila* retina, with photoreceptors highlighted in violet and the pigmented glial cells (PGCs) in grey. PR: photoreceptor**;** R: rhabdomere; CB: cell body.

B–K   Transmission electron microscopy (TEM) images of 5-week-old adult retinas showing the overall structure of groups of ommatidia in (B) wild type; (C) $ADAM17^{-/-}$ mutant; (D) $ADAM17^{-/Deficiency}$; (E, F) RNAi against *ADAM17* and control (*lacZ*), expressed throughout the retina; (G, H) RNAi against *ADAM17* and control (*lacZ*), expressed in the neurons; (I, J) RNAi against *ADAM17* and control (*lacZ*), expressed in the pigmented glial cells; and (K) a $kuz^{-/-}$ mutant.

L     Quantitation of the percentage of normal, abnormal and missing rhabdomeres from TEM images for the genotypes above, expressed as a stacked bar chart; $n$ = 180 ommatidia from 3 different fly retinas for each.

M     Whole-mount retina stained with anti-ADAM17 (green) and phalloidin to mark the photoreceptor membranes (red).

N     Line intensity profiles delineating expression of ADAM17 and phalloidin across individual ommatidia, $n$ = 10 fly retinas.

Data information: Scale bars: 10 μm.

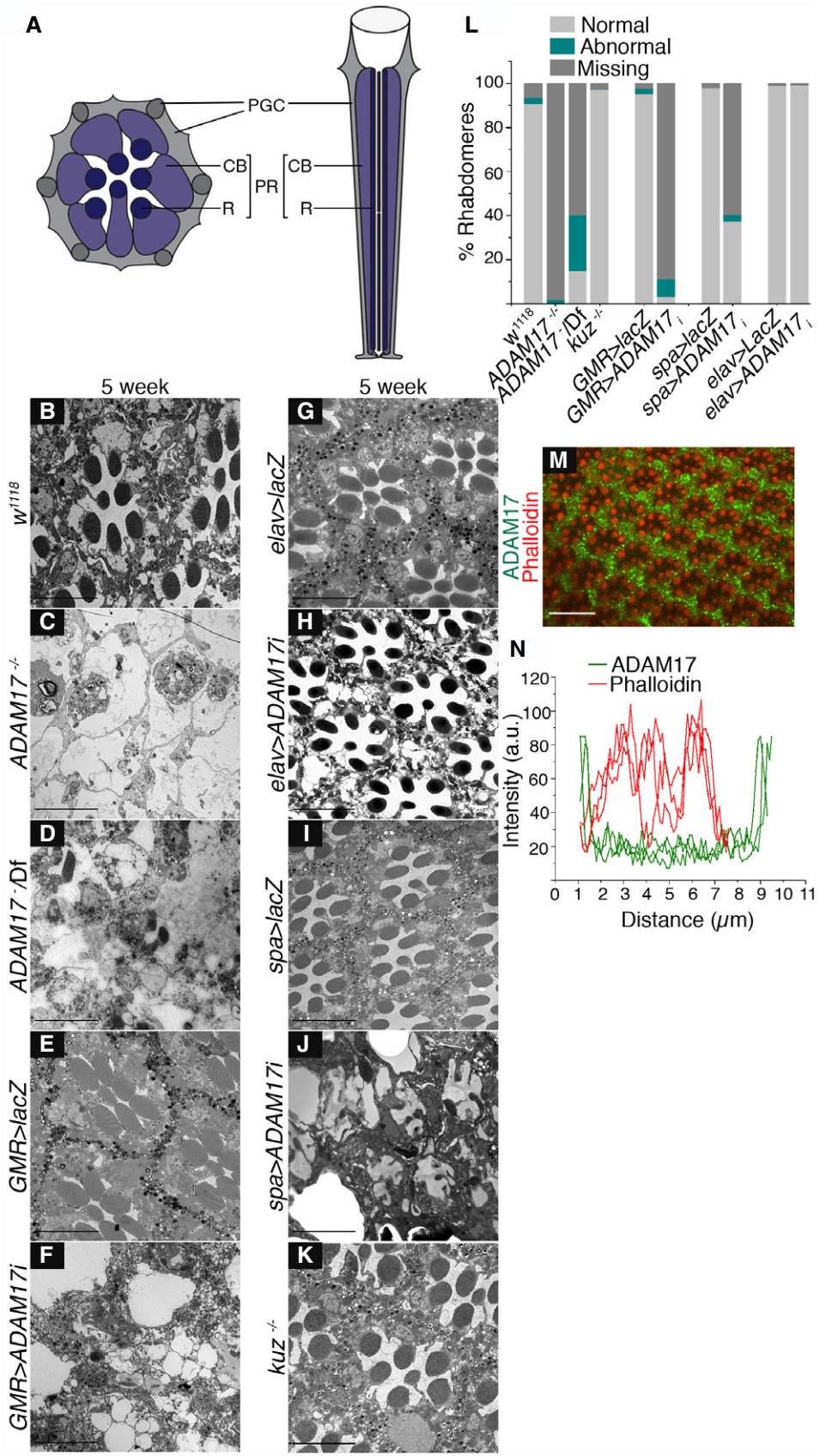

Figure 1.

the *Drosophila* orthologue of ADAM10 (Qi *et al*, 1999). *kuz*$^{-/-}$ retinas showed no signs of degeneration (Fig 1K and L).

The degeneration we observed was progressive with age. Ommatidia in the retinas of 1-day-old *ADAM17* mutant flies were mostly intact (Fig 2A and B), although they did display enlarged PGCs, indicating some abnormalities. Retinas of 2-, 3- and 4-week-old flies showed progressively more severe phenotypes, with only few identifiable neurons or glia seen in *ADAM17* mutants from about 3 weeks of age (Fig EV1E–K). Collectively, these observations imply that ADAM17 in the glial cells of the adult retina protects both neurons and glia from age-dependent degeneration.

### Retinal degeneration of ADAM17 mutant flies is associated with LD accumulation

To investigate the fundamental cause of the retinal degeneration, we looked in more detail for the earliest detectable phenotype. We saw no visible exterior eye defects, suggesting that there was no defect in eye development (Appendix Fig S1A and B). However, ultrastructural investigation by transmission electron microscopy revealed that PGCs of 1-day-old *ADAM17*$^{-/-}$ and *ADAM17*$^-$/Df flies were enlarged and had an accumulation of lipid droplet-like structures; these were rare in wild-type retinas (Fig 2A–C). To confirm the identity of these structures, we labelled 1-day-old retinas with BODIPY 493/503 and FM 4-64FX dyes, which, respectively, label neutral lipids and cellular membranes. There was a striking accumulation of BODIPY-positive LDs in both *ADAM17*$^{-/-}$ and *ADAM17*$^-$/Df retinas (Fig 2D–F and K). In order to quantify and compare lipid accumulation between conditions, we used image analysis to measure LD number, size and a combined measure of "integrated total lipid" (see Materials and Methods). In comparison with wild type, we observed a clear increase in the average size of LDs in *ADAM17*$^{-/-}$ mutants (Fig 2L), and an even more pronounced increase in integrated total lipid (Fig 2K). To strengthen the identification of these structures as LDs, we co-stained wild-type and *ADAM17*$^{-/-}$ adult retinas with BODIPY and lysotracker to check that they were not in fact lysosomal. There was no co-localisation between the two markers in either of the two genotypes (Pearson's correlation coefficient < 0.02; Appendix Fig S1C–E). We also assayed the levels of lipid storage droplet-2 (LSD2), a protein known to associate with lipid droplets, through Western blot (Welte *et al*, 2005). Consistent with our conclusion, LSD2 was upregulated

in *ADAM17*$^{-/-}$ head lysates when compared to wild type (Appendix Fig S1F and G).

LD accumulation was also caused by knockdown of ADAM17 with *Actin-GAL4*, which drives expression in both neurons and glial cells (Fig 2G, K and L). As with the degeneration phenotype, knockdown of ADAM17 specifically in PGCs, but not in neurons, also led to LD accumulation (Fig 2H, I, K and L). Also consistent with the degeneration phenotype, we showed that glial-specific expression of ADAM17 rescued the LD phenotype of the *ADAM17*$^{-/-}$ mutant (Fig EV2A–E) and that loss of the ADAM10 homologue, Kuzbanian, caused no accumulation of LDs in the retina (Fig 2J–L).

Increase in LD amount and size was accompanied by increased expression of lipogenic genes. We compared by qPCR the transcript levels of acetyl coA carboxylase (ACC) and fatty acid synthase 1 (FASN1), both enzymes in the lipid biosynthetic pathway. Both genes were upregulated in *ADAM17*$^{-/-}$ fly head lysates (Fig 2M and N). Consistent with all other data, when knocking down ADAM17 specifically in neurons or glia, we found that lipid synthesis genes only responded to glial ADAM17 loss (Fig EV2F and G). We also looked for defects earlier in retinal development. There was no upregulation of lipogenic gene transcripts, or in the numbers of LDs, in the larval brain, or eye or wing imaginal discs of *ADAM17*$^{-/-}$ mutants (Fig EV3A–H). Similarly, we observed no difference in total LD numbers between wild-type and *ADAM17*$^{-/-}$ pupal retinas (Fig EV3I–K). Furthermore, there was a sharp rise in ADAM17 transcript and protein levels between 40 h pupae and 1-day-old adults (Fig EV3L and M). Together, these results strongly suggest that the defects associated with a loss of ADAM17 arise in adulthood (or possibly very late pupal stages), rather than being caused by earlier developmental defects.

In summary, glial LD accumulation and upregulation of lipogenic genes in *ADAM17*$^{-/-}$ mutants mirrored, but preceded, cell degeneration in all experimental contexts we tested.

### Disrupting LD accumulation rescues degeneration in *ADAM17*$^{-/-}$ mutants

There is a growing link between LD accumulation and the onset of neuronal degeneration (see, for example, Liu *et al*, 2015; Van Den Brink *et al*, 2018). When we measured the temporal pattern of LD accumulation in *ADAM17*$^{-/-}$ adult retinas, we observed a sharp decline in the numbers of LDs with age, starting at around 1 week,

---

**Figure 2. Abnormal LD accumulation in young adult retinas upon a loss of ADAM17 in glial cells.**

A–C    TEM images of 1-day-old adult retinas showing the overall structure of clusters of ommatidia (A1, B1, C1), or a single ommatidium (A2, B2, C2) in wild type (A1, A2), *ADAM17*$^{-/-}$ mutant (B1, B2) or *ADAM17*$^-$/Df (C1, C2). Red arrows show lipid droplets in PGCs, and green shading highlights PGCs.

D–J    Fluorescent images of 1-day-old fly retinas stained with BODIPY (green) and FM dye (red) to mark lipid droplets and the photoreceptor membranes, respectively; (D) wild type; (E) *ADAM17*$^{-/-}$ mutant; (F) *ADAM17*$^-$/$^{Deficiency}$; (G) knockdown of ADAM17 throughout the retina; (H) knockdown in glial cells; (I) knockdown in neurons; and (J) *kuz*$^{-/-}$; *n* = 10 fly retinas.

K, L    Quantitation of the BODIPY signal shown as integrated total lipid and normalised LD size of lipid droplets for the genotypes in (D–I). The box end points are the upper (75%) and lower (25%) quartiles, the whiskers define the maximum 95th percentile and minimum 5th percentile values, respectively, the central band is the median, and the square is the mean. Data were analysed using the Kruskal–Wallis test followed by Dunn's test for *post hoc* analysis for significance due to unequal sample sizes. ***$P$ < 0.001, **$P$ < 0.01.

M, N    Q-PCR analysis of mRNA transcripts of lipogenic genes -Acetyl CoA carboxylase (ACC) and Fatty Acid Synthase (FASN) from heads of wild-type and *ADAM17*$^{-/-}$ mutants; *n* = 4 and *n* = 3 biological replicates for M and N, respectively, with 3 technical replicates for each genotype. The box end points are the upper (75%) and lower (25%) quartiles, the whiskers define the maximum 95th percentile and minimum 5th percentile values, respectively, the central band is the median, and the square is the mean. Data were quantified for significance using Student's *t* test. ***$P$ < 0.001, **$P$ < 0.01.

Data information: Scale bars for A2, B2 and C2: 2 μm and 10 μm for all other panels.

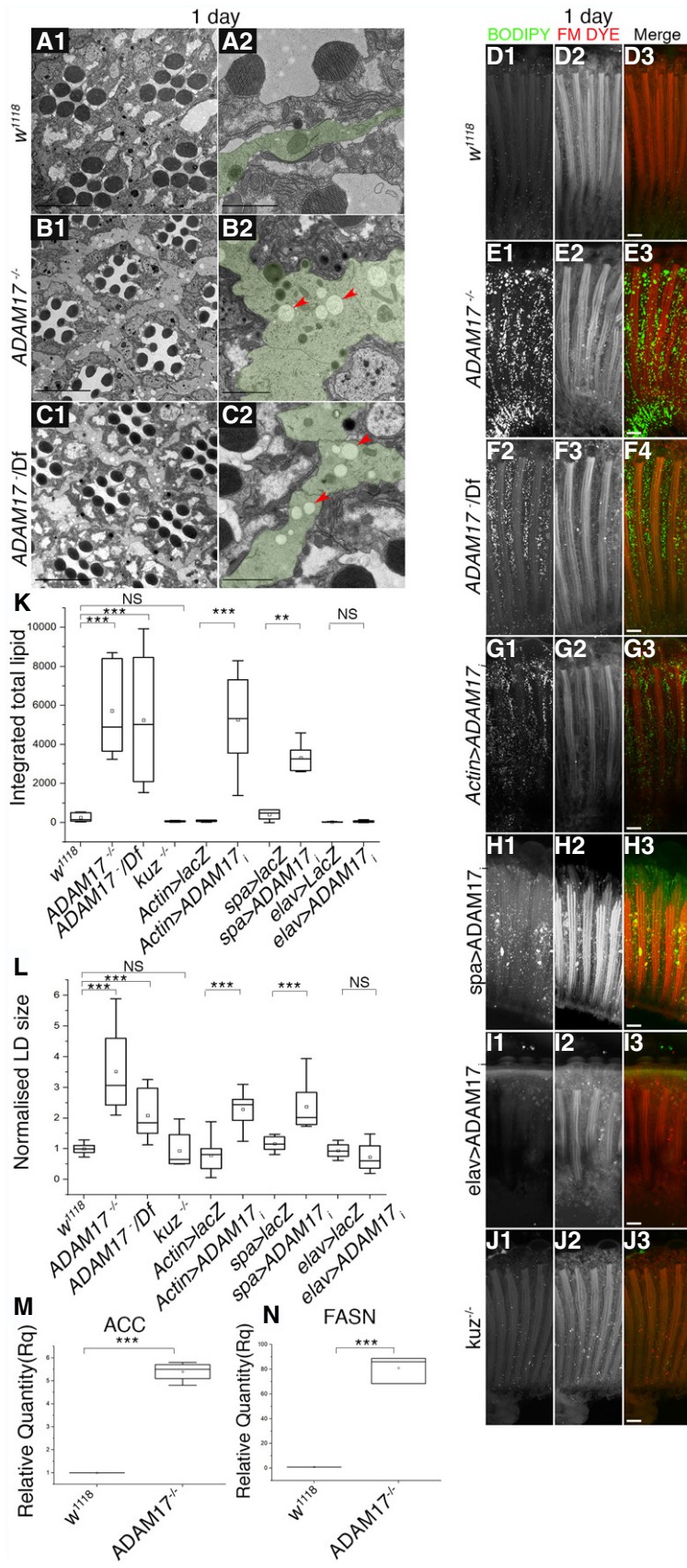

Figure 2.

with their almost complete disappearance by about 2 weeks after eclosion (Appendix Fig S2A–H). This corresponds to the time when degeneration in the $ADAM17^{-/-}$ retinas was becoming prominent (Fig EV1E, F and K). To explore a potential functional link between LDs and cell degeneration, we asked whether degeneration depends on prior LD accumulation. We expressed the LD-associated lipase,

Brummer (Bmm), a homologue of human adipocyte triglyceride lipase, either in PGCs or neurons using the same drivers as above, which are expressed from early in development. In PGCs, this continuously elevated lipase activity led to a striking reduction of LD accumulation in $ADAM17^{-/-}$ retinas (Fig 3A–C and E–G). In contrast, lipase expressed specifically in neurons had no effect on

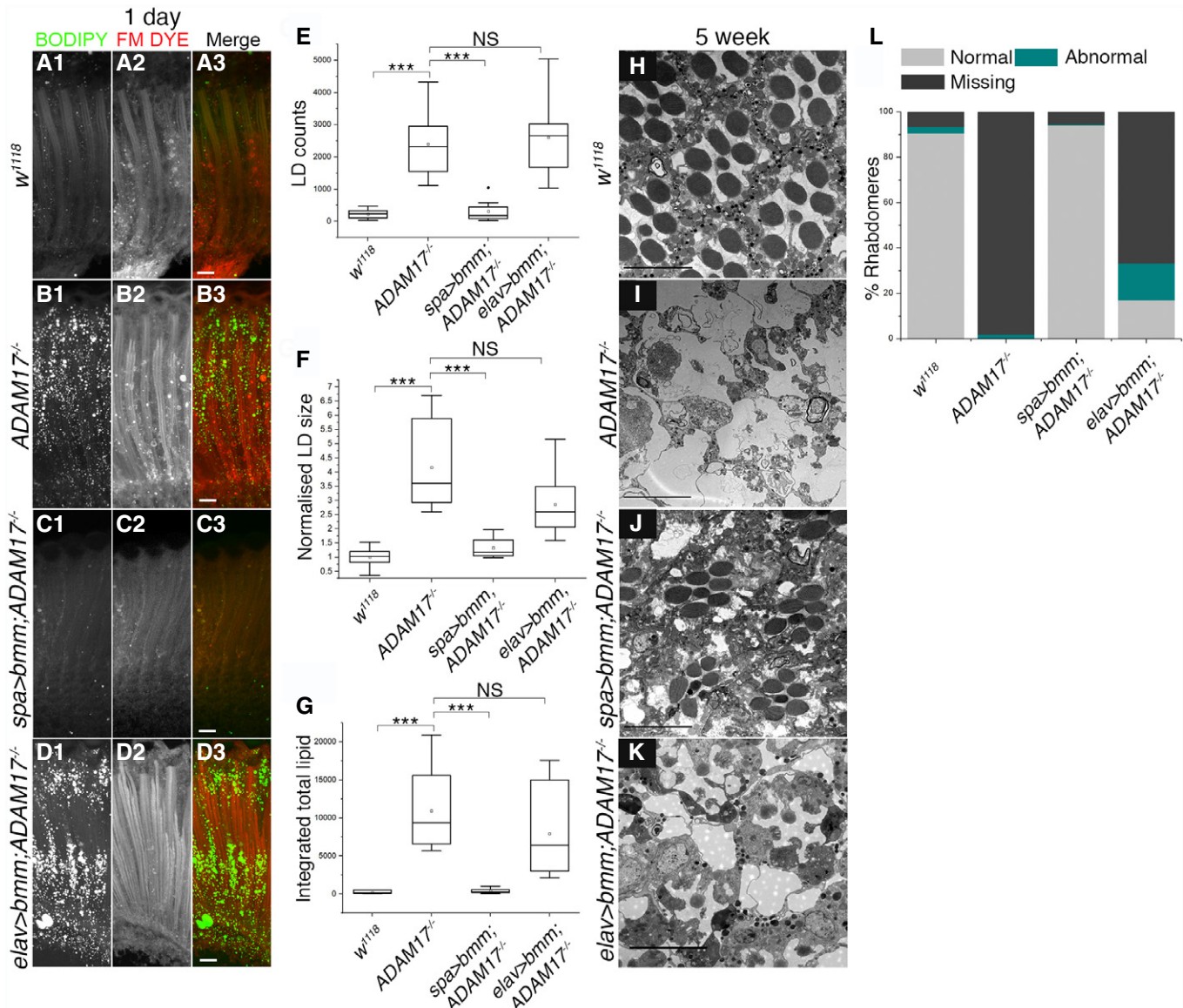

**Figure 3. Overexpressing lipase in $ADAM17^{-/-}$ mutant retinas rescues both lipid droplet accumulation and retinal degeneration.**

A–D  Fluorescent images of 1-day-old retinas, stained with BODIPY (green) and FM dye (red) to stain lipid droplets and the photoreceptor membranes, respectively; (A) wild type; (B) $ADAM17^{-/-}$ mutant; (C) overexpression of lipase in glial cells of $ADAM17^{-/-}$ mutant; (D) overexpression of lipase in neurons of $ADAM17^{-/-}$ mutant.

E–G  Quantitation of the BODIPY signal shown as (E) LD numbers; (F) normalised LD size; (G) integrated total lipid, for the genotypes mentioned above, $n$ = 10 for each genotype. The box end points are the upper (75%) and lower (25%) quartiles, the whiskers define the maximum 95th percentile and minimum 5th percentile values, respectively, the central band is the median, the square is the mean, and the diamond an outlier.

H–K  TEM images of 5-week-old adult retinas corresponding to (H) wild type; (I) $ADAM17^{-/-}$ mutant; (J) glial overexpression of lipase in wild-type glial cells; (K) neuronal overexpression of lipase in the $ADAM17^{-/-}$ mutant.

L  Quantitation of the percentage of normal, abnormal and missing rhabdomeres observed in the TEM images for the genotypes mentioned above, presented as a stacked graph plot; $n$ = 180 ommatidia from 3 different fly retinas for each.

Data information: Data were quantified for significance using Student's $t$ test. ***$P$ < 0.001, **$P$ < 0.01. Scale bars: 10 μm.

the LD accumulation phenotype of ADAM17 loss (Fig 3D–G). Consistent with these results, upon ageing, the flies in which LD accumulation was prevented by glial lipase expression showed almost complete rescue of retinal degeneration; there was much less rescue of degeneration when lipase was overexpressed in neurons (Fig 3H–L). We conclude that prior LD accumulation in PGCs is causally linked to the degeneration induced by loss of ADAM17 in the adult retina; also that the primary source of the abnormally accumulating lipids in these mutants are the glial cells themselves.

### *Drosophila* ADAM17 has metalloprotease activity

Is *Drosophila* ADAM17 an active protease like its mammalian counterpart? It has been predicted to be, but there has been no direct demonstration of its ability to cleave substrates. Alignment of *Drosophila* and human ADAM17 revealed that the N-terminal pro-domain is very different between species and that fly ADAM17 has a shorter C-terminus compared to its human counterpart (Appendix Fig S3A). Conversely, two essential functional domains, the catalytic site and the CANDIS juxtamembrane domain, are well conserved (Appendix Fig S3A), suggesting that the *Drosophila* enzyme may also have metalloprotease activity. To test this, we co-expressed *Drosophila* ADAM17 together with alkaline phosphatase tagged Eiger, the *Drosophila* homologue of TNF (Igaki *et al*, 2002) in *Drosophila* S2R+ cells. Wild-type ADAM17, but not a mutant in which the protease catalytic site was defective (dADAM17HE), was able to proteolytically shed Eiger from S2R+ cells in response to stimulation by the ADAM17 activator PMA (Sahin *et al*, 2004). Shedding was inhibited by the broad-spectrum ADAM protease inhibitor TAPI-1 (Fig 4A). We also showed that *Drosophila* ADAM17 cleaved a canonical mammalian ADAM17 substrate, the interleukin1 receptor IL-1R$_{II}$ (Lorenzen *et al*, 2016; Appendix Fig S3B). Since TAPI-1 could inhibit *Drosophila* ADAM17, we used it to test whether the ADAM17 requirement in flies was dependent on its proteolytic activity. Wild-type *Drosophila* reared on TAPI-1 containing food showed

LD accumulation, as seen in $ADAM17^{-/-}$ mutants (Fig 4B–D), suggesting that the proteolytic activity of ADAM17 is indeed necessary to prevent abnormal LD accumulation. This does not rule out the possibility that other proteases sensitive to TAPI-1 could also contribute.

These data lead us to conclude that *Drosophila* ADAM17 shares overlapping proteolytic activity with conventional mammalian ADAM metalloproteases and that, as in mammals, Eiger/TNF is indeed a substrate.

### TNF pathway inhibition leads to LD accumulation and degeneration

As well as Eiger/TNF, *Drosophila* has two TNF receptor homologues, Grindelwald and Wengen (Kanda *et al*, 2002; Kauppila *et al*, 2003; Andersen *et al*, 2015). There has been some disagreement over the function of TNF signalling in flies, but among other roles, Eiger can trigger cell death (Igaki *et al*, 2002; Narasimamurthy *et al*, 2009). In another context, Eiger acts as an adipokine, released by fat body (adipocyte-like) cells and received in the brain by the Grindelwald receptor, stimulating the release of insulin (Agrawal *et al*, 2016). Having shown that Eiger can be shed by ADAM17, we tested whether TNF signalling mediates the function of ADAM17 in protecting retinal cells from LD accumulation and degeneration. Loss of either Eiger or Grindelwald caused an increase in LDs in the retina (Fig 4E–G and I). This was not the case with loss of Wengen, which was indistinguishable from wild type (Fig 4E, H and I). LD accumulation in *eiger* and *grindelwald* mutants was noticeably slower than in *ADAM17* mutants, visible from day 1 in about 50% of retinas but reaching full penetrance and maximum levels at about 1 week, instead of 1 day (Fig 4E–G and I). Both *eiger* and *grindelwald* mutants also showed elevated lipogenic transcripts (*ACC* and *FASN1*), although again the phenotype had a delayed onset compared to *ADAM17* mutants (Fig 4J and K). Both mutants also displayed signs of retinal degeneration by 2 weeks of age, and by

---

**Figure 4. Loss of either Eiger or Grindelwald, but not Wengen, leads to abnormal LD accumulation and age-associated retinal degeneration.** ▶

A    Alkaline phosphatase (AP)-shedding assays performed for Eiger in S2R+ cells expressing either GFP, *Drosophila* ADAM17, or activity dead mutants of *Drosophila* ADAM17, in the presence of either PMA, or PMA and TAPI-1 (DMSO is used as control; see Materials and Methods); *n* = 5. The box end points are the upper (75%) and lower (25%) quartiles, the whiskers define the maximum 95th percentile and minimum 5th percentile values, respectively, the central band is the median, and the square is the mean.

B, C   Fluorescent images of retinas from 1-day-old flies grown on either DMSO or TAPI-1, stained with BODIPY (green) and FM dye (red) to mark lipid droplets and photoreceptor membranes, respectively.

D    Integrated total lipid counts from retinas corresponding to flies reared either on DMSO or TAPI-1 starting from larval stages. *n* = 10 fly retinas. The box end points are the upper (75%) and lower (25%) quartiles, the whiskers define the maximum 95th percentile and minimum 5th percentile values, respectively, the central band is the median, and the square is the mean.

E–H   Fluorescent images of 1-week-old adult fly retinas, stained with BODIPY (green) and FM dye (red) to mark lipid droplets and photoreceptor membranes respectively: (E) wild type; (F) *eiger*$^{-/-}$; (G) *grindelwald*$^{-/-}$; (H) *wengen*$^{-/-}$.

I    Quantitation of the BODIPY signal shown as integrated total lipid measurements for the genotypes mentioned above. *n* = 10 fly retinas. The box end points represent the upper (75%) and lower (25%) quartiles, the whiskers define the maximum 95th percentile and minimum 5th percentile values, respectively, the central band is the median, the square is the mean, and the diamond an outlier.

J, K   mRNA level transcripts of ACC and FASN measured from head lysates of 1-week-old wild type, *eiger*$^{-/-}$ and *grindelwald*$^{-/-}$. *n* = 3 biological replicates, with 3 technical replicates for each genotype. The box end points represent the maximum and minimum values, respectively, the central band is the median, and the square is the mean.

L–O   TEM images of 2-week-old retinas corresponding to (L) control; (M) *eiger*$^{-/-}$; (N) *grindelwald*$^{-/-}$; (O) *wengen*$^{-/-}$. *n* = 180 ommatidia from 3 different fly retinas for each.

P    Quantitation of the percentage of normal, abnormal and missing rhabdomeres from 2-week-old retinas observed in TEM, corresponding to the genotypes mentioned above; *n* = 180 ommatidia from 3 different fly retinas for each.

Data information: Data were quantified for significance using Student's *t* test. ***P < 0.001, **P < 0.01. Scale bars: 10 μm.

---

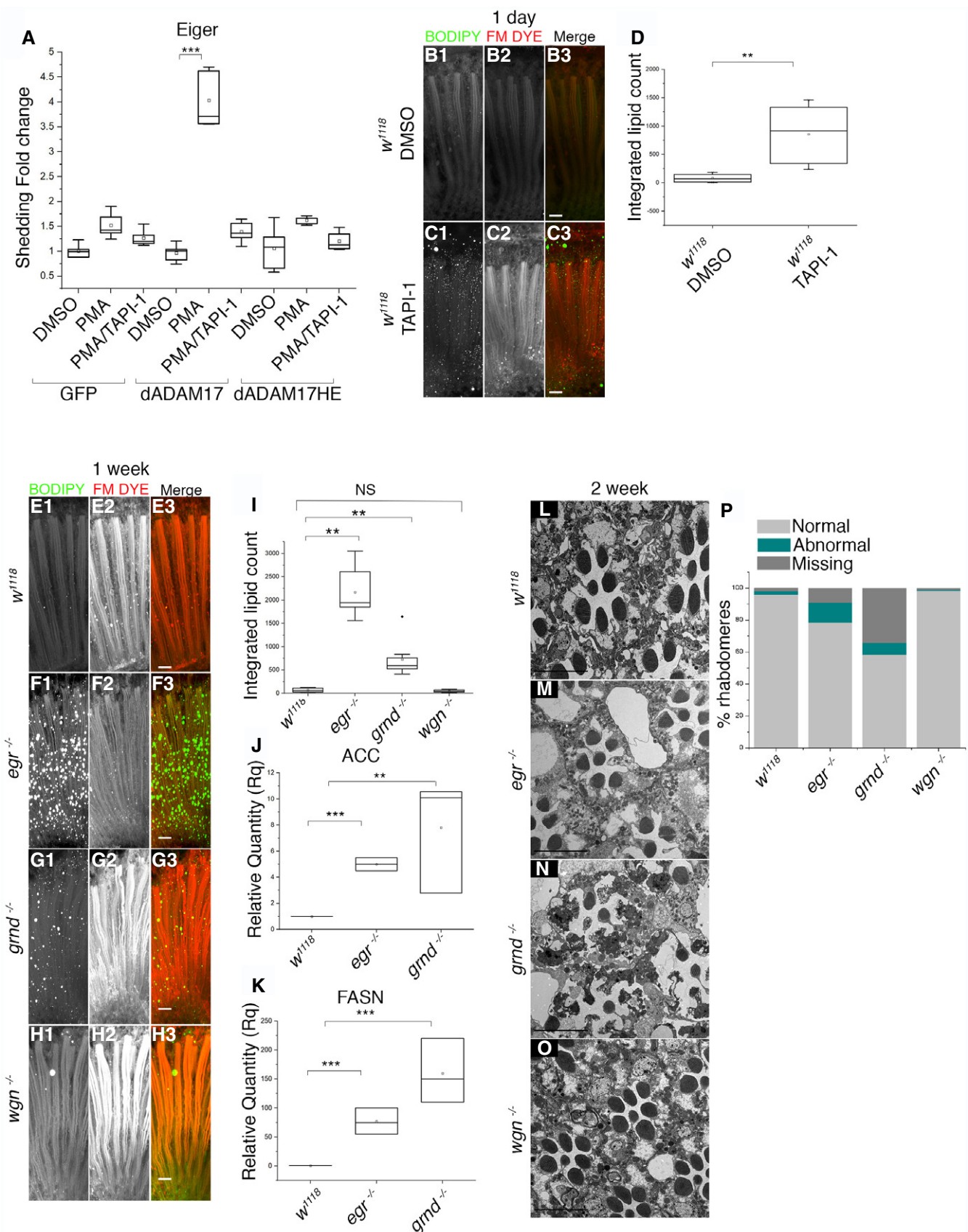

**Figure 4.**

5 weeks, full degeneration was observed (Figs 4L–N and P, and EV4N–Q). No retinal degeneration was seen with *wengen* mutants (Fig 4O and P).

Since Eiger is a secreted ligand and Grindelwald a receptor, they have the capacity to act in different cells. We therefore investigated in which cell types each was needed to protect cells from degeneration. Using cell-type specific RNAi, we showed that both Eiger and Grindelwald are needed specifically in glia: loss of either in PGCs, but not neurons, caused LD accumulation (Fig EV4A–G), as well as the upregulation of lipogenic genes (Fig EV4H and I). Knockdown of either Eiger or Grindelwald in neurons alone caused no abnormal accumulation of LDs (Fig EV4G, J and L). This suggests no requirement for Eiger or Grindelwald in neurons, at least with respect to LDs. Loss of Wengen, either in glia or neurons, had no significant effect on LD accumulation or expression of lipogenic transcripts (Figs 4H and EV4D–I).

Overall, these results indicate that ADAM17 protects retinal cells from LD accumulation and degeneration by shedding Eiger and subsequent activation of the TNF receptor Grindelwald, but not Wengen. Moreover, our data support a model where the ligand and the receptor are both required in glial cells.

## LD accumulation in *ADAM17* mutants depends on ROS and elevated JNK signalling

Defective mitochondria in photoreceptor neurons cause transient accumulation of LDs in neighbouring glial cells, which depends on the accumulation of reactive oxygen species (ROS) and stress signalling by the c-Jun N-terminal kinase (JNK) pathway in the photoreceptors (Liu *et al*, 2015, 2017). We therefore asked whether the retinal degeneration caused by glial ADAM17 loss also depends on ROS. We first grew larvae on food supplemented with the antioxidant N-acetyl cysteine amide (AD4), a ROS quencher that can cross the blood–brain barrier (Amer *et al*, 2008; Schimel *et al*, 2011). AD4 treatment rescued both LD accumulation (Fig 5A–E) and age-associated degeneration of $ADAM17^{-/-}$ retinas (Fig 5F and J), suggesting that ROS contribute to the $ADAM17^{-/-}$ phenotype. To validate this idea and identify the ROS source, we genetically quenched ROS in glia or neurons by cell-type-specific expression of super oxide dismutase 2 (SOD2), a mitochondrial enzyme that destroys mitochondrially generated ROS (Kirby *et al*, 2002). Quenching mitochondrial ROS specifically in glia of $ADAM17^{-/-}$ retinas fully rescued LD accumulation, overexpression of lipogenic transcripts, and degeneration (Figs 5K, L, O and P–R, and EV5D and E). Significantly lower rescue was seen with glial-specific overexpression of SOD1, a cytosolic ROS quencher (Figs 5M and O, and EV5D and E). We also did the same experiment with catalase, another cytosolic ROS quenching enzyme. Again in contrast to mitochondrial SOD2, glial expression of catalase had little effect on LD accumulation or the induction of lipogenic genes (Fig EV5A–C), supporting the conclusion that the phenotypes caused by loss of ADAM17 depend mainly on mitochondrially generated ROS. Interestingly, neuronal-specific SOD2 also caused partial rescue of LD accumulation, less complete than SOD2 expressed in glia, but significant nonetheless (Fig 5N and O). Overexpressing SOD2 in glial cells of the $ADAM17^{-/-}$ retinas also led to a significant rescue of retinal degeneration (Fig 5P–R). In further support for the accumulation of ROS in $ADAM17^{-/-}$ retinas, we performed an Oxyblot experiment to measure the extent of protein oxidation, a readout commonly used to measure cellular ROS levels (Chen *et al*, 2019). This showed a clear increase in $ADAM17^{-/-}$ mutant retinas compared to wild type (Fig 5S and T).

Our observation that ROS quenching in neurons can partially rescue $ADAM17^{-/-}$ implies that although the protective function of ADAM17 is strictly glial cell specific, and glial-generated ROS are essential to trigger LD accumulation and degeneration, neuronal ROS also contributes significantly to the phenotype. To validate and

---

**Figure 5.** Reducing mitochondrial ROS levels in $ADAM17^{-/-}$ mutants leads to a rescue of the LD accumulation and retinal degeneration. ▶

A–D Fluorescent images of 1-day-old adult fly retinas, stained with BODIPY (green) and FM dye (red) to mark lipid droplets and photoreceptor membranes respectively: (A, B) wild type; (C, D) $ADAM17^{-/-}$ mutants, reared either on DMSO or AD4.

E Integrated total lipid counts from retinas corresponding to the genotypes and treatments mentioned above; n = 10 for each. The box end points represent the upper (75%) and lower (25%) quartiles, the whiskers define the maximum 95th percentile and minimum 5th percentile values, respectively, the central band is the median, and the square is the mean. Data were analysed using the Kruskal–Wallis test followed by Dunn's test for *post hoc* analysis for significance due to unequal sample sizes. ***$P < 0.001$.

F–I TEM images of 2-week-old retinas corresponding to either wild-type or $ADAM17^{-/-}$ mutants grown on either DMSO or AD4.

J Percentage of normal, abnormal or missing rhabdomeres from retinas corresponding to the genotypes and treatments mentioned above. n = 180 ommatidia from 3 different fly retinas for each.

K–N Fluorescent images of 1-day-old adult fly retinas, stained with BODIPY (green) and FM dye (red) to mark lipid droplets and photoreceptor membranes, respectively: (K) $ADAM17^{-/-}$; (L) glial-specific SOD2 overexpression in an $ADAM17^{-/-}$ mutant; (M) glial-specific SOD1 overexpression in an $ADAM17^{-/-}$ mutant; (N) neuronal-specific SOD2 overexpression in $ADAM17^{-/-}$ mutant.

O Quantitation of the BODIPY signal shown as integrated total lipid for the genotypes mentioned above, n = 10 for each. The box end points are the upper (75%) and lower (25%) quartiles, the whiskers define the maximum 95th percentile and minimum 5th percentile values, respectively, the central band is the median, the square is the mean, and the diamond an outlier. Data were analysed using the Kruskal–Wallis test followed by Dunn's test for *post hoc* analysis for significance due to unequal sample sizes. ***$P < 0.001$, **$P < 0.01$, *$P < 0.05$.

P, Q TEM analysis of 5-week-old retinas corresponding to either $ADAM17^{-/-}$ mutants or SOD2 overexpression in an $ADAM17^{-/-}$ mutant.

R Percentage of normal, abnormal and missing rhabdomeres, as compared to retinas overexpressing glial-specific SOD2 in an $ADAM17^{-/-}$ mutant background observed with TEM; n = 180 ommatidia from 3 different fly retinas for each.

S Oxyblot analysis of oxidative modification of proteins from retina lysates of wild type and $ADAM17^{-/-}$.

T Analysis of Oxyblot intensity normalised to its respective actin controls for each genotype. n = 3 biological replicates, with 3 technical replicates for each genotype. The box end points represent the maximum and minimum values, respectively, the central band is the median, and the square is the mean. Data were quantified for significance using Student's t test. **$P < 0.01$.

Data information: Scale bars: 10 μm.

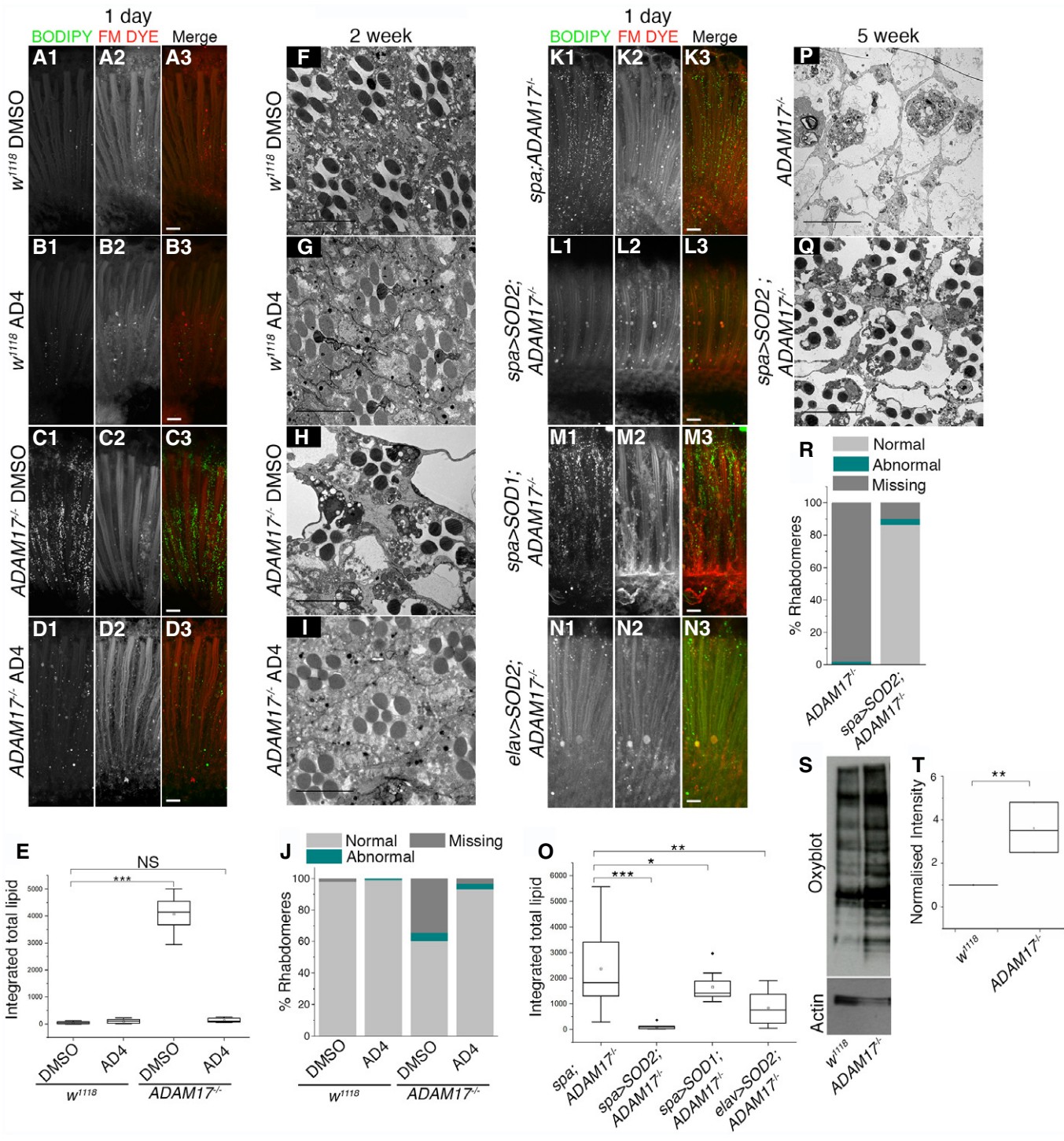

**Figure 5.**

explore this involvement of neurons, we raised *ADAM17⁻/⁻* flies in total darkness, thereby preventing physiological ROS generation by normal photoreceptor activity. These flies exhibited almost complete rescue of degeneration and a significant, though not complete rescue of LD accumulation (Fig 6A–H). We conclude that the toxicity that leads to retinal degeneration in *ADAM17⁻/⁻* mutants is contributed to by ROS from neighbouring photoreceptors as well as

ROS generated in the glia. Moreover, our data imply that the function of ADAM17 and TNF signalling in glia helps protect cells from the cumulative damage of ROS generated by normal light-induced neuronal activity. There is evidence that the Nrf2-Keap1 pathway can link TNF signalling to ROS production (Shanmugam *et al*, 2016), so we examined some of the transcriptional targets of the Nrf2-Keap1 pathway in *ADAM17⁻/⁻* mutants. SOD2, catalase and

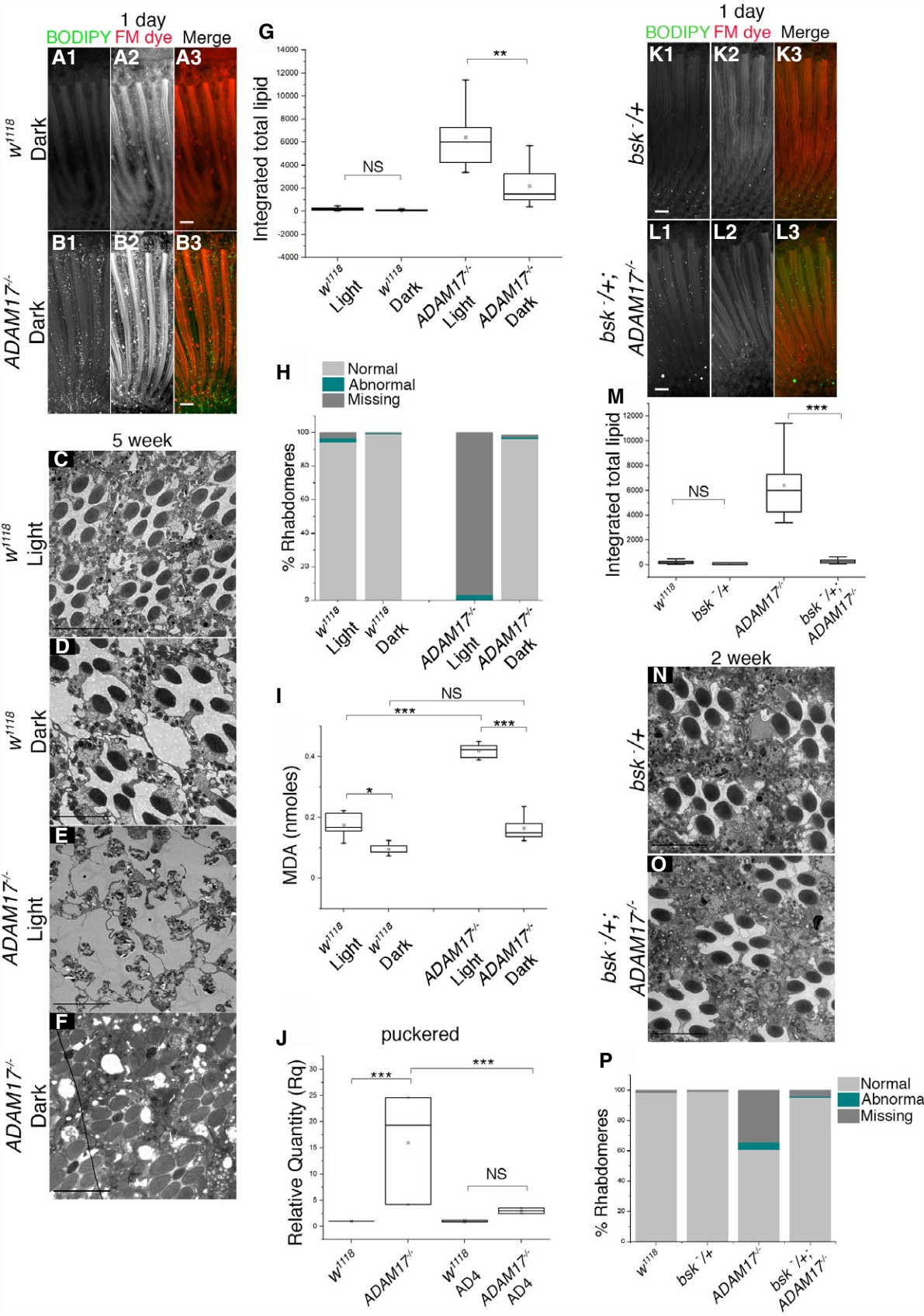

**Figure 6.**

◀

**Figure 6.   Activity dependence and role of JNK in driving LD accumulation and retinal degeneration in *ADAM17*$^{-/-}$ mutants.**

A, B    Fluorescent images of 1-day-old adult fly retinas, stained with BODIPY (green) and FM dye (red) to mark lipid droplets and photoreceptor membranes, respectively: (A) wild type; (B) *ADAM17*$^{-/-}$, both reared in the dark.

C–F    TEM images of 4-week-old retinas: (C, D) wild-type or (E, F) *ADAM17*$^{-/-}$ flies reared from first larval instar either in total light (C, E) or total dark (D, F).

G       Quantitation of the BODIPY signal shown as integrated total lipid from 1-day-old retinas corresponding to the genotypes described in A, B, all reared in the dark. *n* = 10 for each. The box end points represent the upper (75%) and lower (25%) quartiles, the whiskers define the maximum 95$^{th}$ percentile and minimum 5$^{th}$ percentile values, respectively, the central band is the median, and the square is the mean.

H       Percentage of normal, abnormal and missing rhabdomeres in the genotypes and treatments mentioned above. *n* = 180 ommatidia from 3 different fly retinas for each.

I       Measurement of MDA levels (in nmoles) from 1-day-old retinas of wild type or *ADAM17*$^{-/-}$, reared either in light or dark. Four biological replicates with 3 technical replicates for each genotype. The box end points represent the upper (75%) and lower (25%) quartiles, the whiskers define the maximum 95$^{th}$ percentile and minimum 5$^{th}$ percentile values, respectively, the central band is the median, and the square is the mean.

J       mRNA levels of the puckered transcripts from retinas of untreated or AD4 treated wild-type and *ADAM17*$^{-/-}$ mutant retinas. *n* = 3 biological replicates, with 3 technical replicates for each genotype. The box end points represent the maximum and minimum values, respectively, the central band is the median, and the square is the mean.

K, L    Fluorescent images of 1-day-old adult fly retinas, stained with BODIPY (green) and FM dye (red) to mark lipid droplets and photoreceptor membranes respectively: (K) heterozygous mutant of *bsk* alone; or (L) in combination with *ADAM17*$^{-/-}$.

M       Quantitation of the BODIPY signal shown as integrated total lipid for the genotypes mentioned in K, L, along with *w1118* and *ADAM17*$^{-/-}$. *n* = 10 flies per genotype. The box end points represent the upper (75%) and lower (25%) quartiles, the whiskers define the maximum 95$^{th}$ percentile and minimum 5$^{th}$ percentile values, respectively, the central band is the median, and the square is the mean.

N, O    TEM images of 2-week-old adult retinas, showing clusters of ommatidia for a heterozygous mutant of *bsk* alone (M) or in combination with *ADAM17*$^{-/-}$ (N); *n* = 3 for each.

P       Percentage of normal, abnormal and missing rhabdomeres for the genotypes mentioned above; *n* = 180 ommatidia from 3 different fly retinas for each.

Data information: Data were quantified for significance using Student's *t* test. **$P$ < 0.01. Scale bars: 10 μm.

glutamate cysteine ligase catalytic subunit (Gclc) were all reduced (Fig EV5F–H). This suggests that loss of ADAM17 leads to a compromised antioxidant response, potentially contributing to elevated levels of ROS. Although quite preliminary, this is an interesting observation that provides possible insight into the downstream cellular pathways that mediate ADAM17's role in protecting retinal cells from degeneration.

The combination of lipid accumulation and elevated ROS in *ADAM17*$^{-/-}$ mutants suggested that toxic peroxidated lipids (Niki, 2009; Gaschler & Stockwell, 2017) might be contributing to retinal degeneration. We therefore measured the levels of malondialdehyde (MDA), a by-product and hallmark of lipid peroxidation (Chen *et al*, 2017). MDA was dramatically increased in *ADAM17*$^{-/-}$ retinas (Fig 6I). When rearing wild-type flies in the dark, we saw a minor reduction of MDA, compared to light-reared controls (Fig 6I). This was expected and reflects the normal level of peroxidated lipids made by ROS generated by photoreceptor activity. More strikingly, the absence of neuronal activity in dark-reared *ADAM17*$^{-/-}$ flies fully rescued the accumulation of toxic peroxidated lipids (Fig 6I). This correlates well with the complete rescue of retinal degeneration seen in dark-reared flies (Fig 6E, F and H), suggesting that peroxidated lipids are indeed the major cause of cell death in *ADAM17*$^{-/-}$ mutants. This rescue of cell death by preventing photoreceptor activity supports the model that ROS contribute to the toxic effects of LDs against which ADAM17 and TNF protect glia.

We investigated whether LD formation and retinal degeneration rely on JNK signalling, a well-characterised mediator of ROS-induced stress (Shen & Liu, 2006). ADAM17 loss caused a significant increase in the expression of *puckered*, a transcriptional target of the JNK pathway (Fig 6J). We also assayed phosphorylated JNK, a directed measure of JNK activity, and found it to be elevated in *ADAM17*$^{-/-}$ retinas (Fig EV5I and J). The functional significance of JNK signalling was demonstrated by the observation that halving the genetic dose of JNK (the *basket* gene in *Drosophila*) strongly suppressed both the LD and degeneration phenotypes caused by

ADAM17 loss (Fig 6K–P). This places JNK activity genetically downstream of ADAM17 loss. We investigated the relationship between ROS and JNK signalling by assaying JNK activity in retinas from flies treated with the antioxidant AD4. Quenching ROS by this means rescued elevated puckered transcript levels (Fig 6J), implying that JNK activation is triggered by ROS.

Overall, these results indicate that ADAM17 in *Drosophila* PGCs protects them from damage caused by ROS accumulation, which leads to the activation of the stress-induced JNK pathways, LD formation, generation of peroxidated lipids, and consequent cellular degeneration.

### Loss of ADAM17 activity drives abnormal LD accumulation and mitochondrial ROS production in human iPSC-derived microglia cells

Our discovery of a new cytoprotective function for ADAM17 in *Drosophila* led us to question whether this role might be conserved in mammals. We therefore used human iPSC-derived microglia-like cells from three different donors and treated them for 24 h with the widely used metalloprotease inhibitors TAPI-1, GW280264X (GW) or GI254023X (GI). TAPI-1 is an inhibitor for ADAM17, but can also to a lesser extent inhibit other ADAMs and matrix metalloproteinases. GW inhibits specifically ADAM17 and ADAM10, while GI is a selective ADAM10 inhibitor (Chalaris *et al*, 2010; Moller-Hackbarth *et al*, 2013). Between them, they are often used to identify ADAM-dependent events and to distinguish ADAM10 from ADAM17 activity. After treatment, cells were labelled with BODIPY 493/503 and FM 4-64FX dyes to mark LDs and membrane respectively. Cells from all three donors treated with TAPI-1 or GW showed an increase in LD size and elevated total lipid, with almost no change in LD number (Fig 7A1–A3, B1–B3, C1–C3, D and E, Appendix Fig S4A). GI treatment, specific for ADAM10, had little effect (Fig 7A4, B4, C4, D and E, and Appendix Fig S4A). These results imply that inhibition of specifically ADAM17 in these human

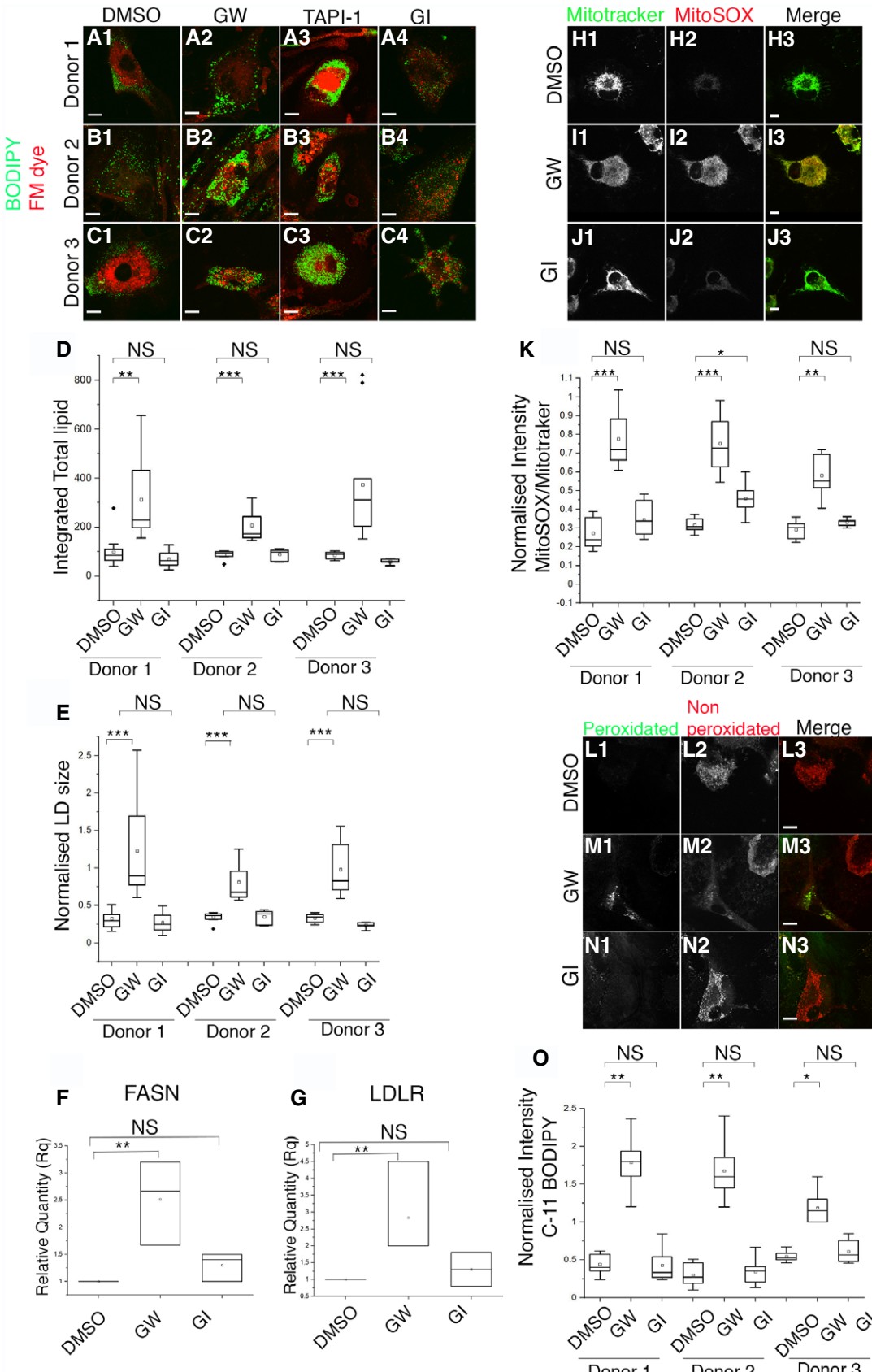

Figure 7.

◄

**Figure 7.  Inhibiting ADAM17 activity in human iPSC-derived microglia cells leads to abnormal LD phenotypes.**

A–C   Fluorescent images of human iPSC-derived microglia cells, labelled with 493/503 BODIPY (green) and FM dye (red) to mark lipids and membranes respectively. iPSC-derived microglia cells were obtained from 3 different donors and treated with either DMSO, GW, TAPI-1 or GI.

D, E   Quantitation of the BODIPY signal shown as integrated total lipid and normalised LD size (normalised to DMSO) for cells corresponding to each treatment and donor mentioned in A–C. $n$ = 15 cells for each treatment. The box end points represent the upper (75%) and lower (25%) quartiles, the whiskers define the maximum 95$^{th}$ percentile and minimum 5$^{th}$ percentile values, respectively, the central band is the median, the square is the mean, and the diamond an outlier. Data were analysed using the Kruskal–Wallis test followed by Dunn's test for *post hoc* analysis for significance due to unequal sample sizes. ***$P < 0.001$.

F, G   Q-PCR analysis of mRNA transcripts of lipogenic genes FASN and LDLR in pooled data obtained from 3 different donors of human iPSC-derived microglia treated with either DMSO, GW or GI for 24 h. $n$ = 3 biological replicates, with 3 technical replicates for each genotype. The box end points represent the maximum and minimum values, respectively, the central band is the median, and the square is the mean.

H–J   Fluorescent images of Mitotracker and MitoSOX labelling of cells from Donor number 2 treated for 24 h with either DMSO, GW or GI.

K      Quantitative measurements of MitoSOX intensity normalised to that of Mitotracker in iPSC-derived microglia cells from three different donors, treated with either DMSO, GW or GI; $n$ = 10 cells for each treatment per donor. The box end points are the upper (75%) and lower (25%) quartiles, the whiskers define the maximum 95$^{th}$ percentile and minimum 5$^{th}$ percentile values, respectively, the central band is the median, and the square is the mean.

L–N   Fluorescent images of C-11 BODIPY treated microglia cells from Donor no. 1, treated for 24 h with either DMSO, GW or GI. These representative images display relative amounts of peroxidated and non-peroxidated lipids.

O      Quantification of BODIPY C-11 staining, shown as ratios of peroxidated versus non-peroxidated lipids, depicted as normalised intensity; $n$ = 10 cells for each treatment per donor. The box end points are the upper (75%) and lower (25%) quartiles, the whiskers define the maximum 95$^{th}$ percentile and minimum 5$^{th}$ percentile values, respectively, the central band is the median, and the square is the mean.

Data information: Apart from the data in panels D and E, all data were quantified for significance using Student's $t$ test. ***$P < 0.001$, **$P < 0.01$, *$P < 0.05$. Scale bars: 10 μm.

cells causes LD defects similar to those observed in flies. We next assayed lipogenic gene expression. GW treatment of all donor cells showed an increase in the mRNA levels of FASN and LDLR, whereas GI inhibition of ADAM10 had no effect, again indicating an ADAM17-specific phenomenon (Fig 7F and G). To extend the analogy between *Drosophila* and human cells, we tested whether inhibiting ADAM17 activity had any effect on mitochondrial ROS levels by live imaging cells with MitoSOX. Consistent with the LD phenotype, inhibition of ADAM17, but not ADAM10, caused striking upregulation of mitochondrial ROS species (Fig 7H–K). Finally, we tested whether, as in flies, loss of ADAM17 in human cells led to an increase in the cytotoxic peroxidated lipids that we propose to be the ultimate product of the pathway. Using the dye C11-BODIPY, which undergoes a spectral shift from 590 to 520 nm when bound to peroxidated lipids, we observed a sharp increase in the numbers of peroxidated LDs, only upon treatment with GW, but not with either DMSO or GI (Fig 7L–O). This result was confirmed by measuring MDA levels in untreated and treated cells. Again, a significant difference was detected in GW and TAPI-1 treated cells, with no effect upon treatment with GI (Appendix Fig S4B), demonstrating that ADAM17 inhibition causes accumulation of peroxidated lipids.

In conclusion, our data show that loss of human ADAM17 leads to the same cellular phenomena in iPSC-derived human microglia as seen in *Drosophila* PGCs: LD accumulation, increased expression of lipogenic genes, elevated mitochondrial ROS and abnormal levels of toxic peroxidated lipids.

## Discussion

We report here a previously unrecognised role of ADAM17 and TNF in protecting *Drosophila* retinal cells from age- and activity-related degeneration. Loss of ADAM17 and TNF signalling in retinal glial cells causes an abnormal accumulation of LDs in young glial cells. These LDs disperse by about 2 weeks after eclosion (middle age for flies), and their loss coincides with the onset of severe glial and neuronal cell death. By 4 weeks of age, no intact glia or neurons

remain. Cell death depends on neuronal activity: retinal degeneration and, to a lesser extent, LD accumulation are rescued in flies reared fully in the dark. We find that LD accumulation does not merely precede, but is actually responsible for subsequent degeneration, because preventing the accumulation of LDs fully rescues cell death. Our data indicate that Eiger/TNF released by ADAM17 acts specifically through the Grindelwald TNF receptor. Loss of ADAM17-mediated TNF signalling also leads to elevated production of mitochondrial ROS in glial cells, causing activation of the JNK pathway and elevated lipogenic gene expression. Together, these changes trigger cell death through the production of toxic peroxidated lipids. Importantly, toxicity is also contributed to by ROS generated by normal activity of neighbouring neurons. Finally, we show that a similar signalling module is conserved in mammalian cells: when ADAM17 is inhibited in human iPSC-derived microglial-like cells, we see the same series of events: LD accumulation, elevated mitochondrial ROS and high levels of toxic peroxidated lipids (Fig 8).

We propose that TNF is an autocrine trophic factor that protects retinal pigmented glial cells from age-related cumulative damage caused by the ROS that are normal by-products of neuronal activity. This ADAM17/TNF protection system is located specifically in retinal glial cells, but its role is to protect both glia and neighbouring neurons. In the absence of this TNF cytoprotective pathway, we see severe early-onset retinal neurodegeneration. Our data imply that cells die by being overwhelmed by toxic peroxidated lipids when abnormal accumulations of LDs disperse. This occurs in *Drosophila* middle age, when LDs stop accumulating and begin to disperse, triggering the cytotoxic phase of the $ADAM17^{-/-}$ phenotype. It is important to emphasise that despite the ADAM17/TNF protection system being located specifically in retinal glial cells, there is neuronal involvement. Not only does TNF indirectly protect against neurodegeneration, but photoreceptor neurons are also significant sources of the ROS that generate the toxic peroxidated lipids in glia. More generally, this work provides a model for investigating more widely the functional links between ageing, cellular stress, lipid droplet accumulation and neurodegeneration. Indeed, in the light of our discovery that the pathway we have discovered in *Drosophila* is

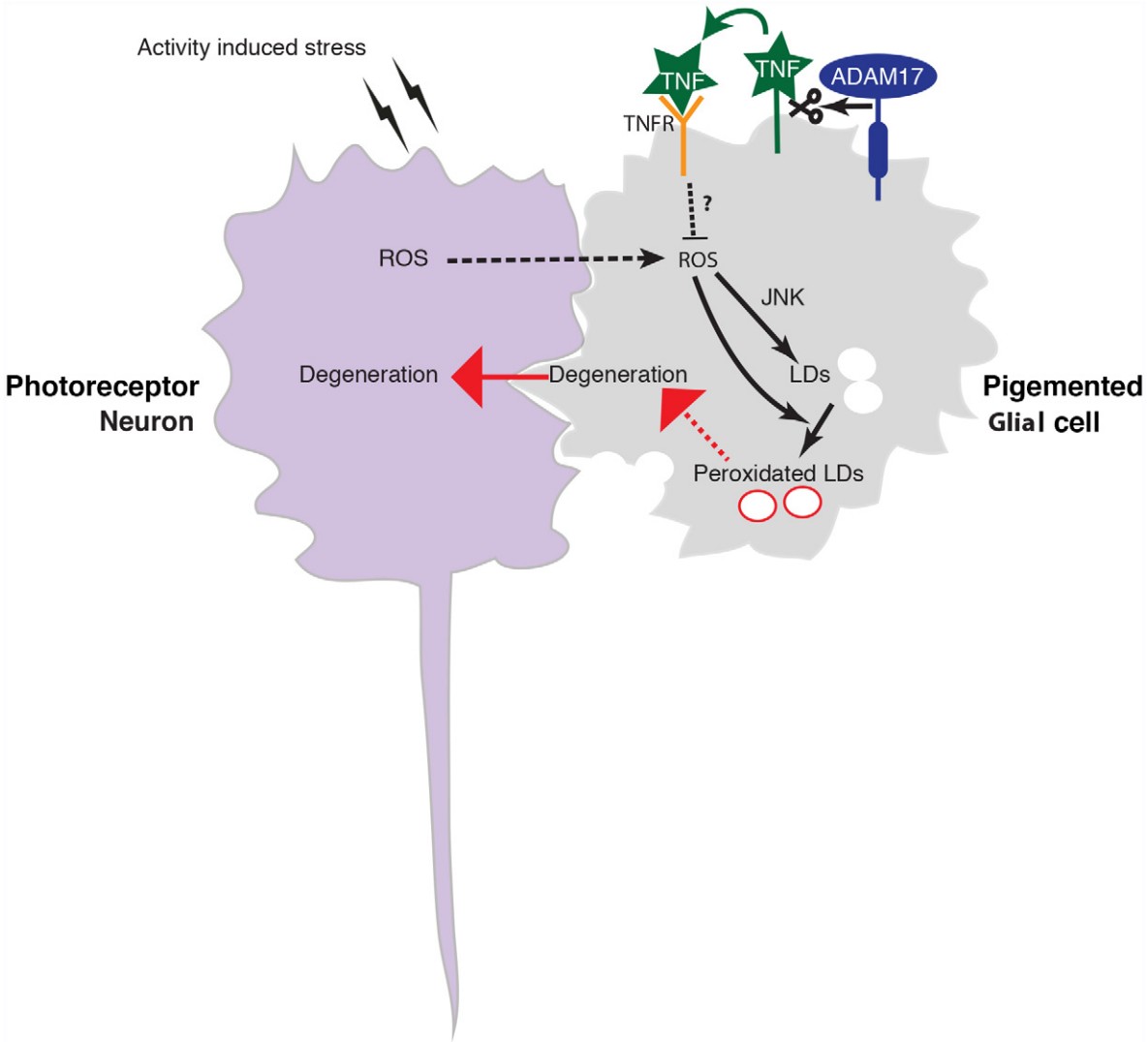

**Figure 8.  Regulation of ROS and peroxidated LDs by ADAM17 and components of the TNF pathway.**

Cleavage of full length TNF/Eiger by ADAM17 leads to the production of soluble TNF/Eiger, which can engage the TNF receptor, Grindelwald. TNF signalling inhibits the production of mitochondrial ROS inside pigmented glial cells, through unknown mechanisms. In the absence of TNF signalling, glial-generated ROS combines with ROS from neighbouring photoreceptor neurons to trigger production of LDs. Upon their later dispersal, the combination of fatty acids and ROS leads to high levels of toxic peroxidated lipids, which are toxic to the glia. Finally, glial degeneration is followed by neuronal degeneration.

conserved in human microglia-like cells, it is significant that lipid droplets have been reported to accumulate in human microglia, cells that are increasingly prominent in the pathology of Alzheimer's disease and other neurodegenerative conditions (Shinozaki *et al*, 2014; Karran & De Strooper, 2016; Bastos *et al*, 2017).

In a *Drosophila* model of neuronal mitochondrionopathies, abnormal neuronal ROS production led to elevated neuronal lipid production, followed by transfer of the lipids to the PGCs, where LDs accumulated (Liu *et al*, 2015, 2017). In that work, lipase expression in neurons suppressed LD accumulation; in contrast, we only observed suppression when lipase was expressed in PGCs, not neurons, suggesting that in the case of ADAM17 mutants, the primary source of accumulating lipids is the glial cells. Despite not being able to detect a role for neuronal lipid production in ADAM17$^{-/-}$ mutants, we do find that photoreceptor

neurons are significant sources of the ROS that generate the toxic peroxidated lipids in glia. Despite these differences between this work and what has been previously reported, a growing body of work points to a close coupling between ROS, the JNK pathway, lipid droplets and cellular degeneration (Liu *et al*, 2015, 2017; Van Den Brink *et al*, 2018), a relationship conserved in mammals (Liu *et al*, 2015; this work). We have not here investigated the involvement of SREBP in mediating LD accumulation caused by ADAM17 loss, but its well-established connection with stress-induced and JNK-mediated lipid synthesis (Liu *et al*, 2015) suggests that it is a likely additional shared component of this conserved regulatory axis.

It has become clear that LDs are much more than simply passive storage vessels for cellular lipids; they have multiple regulatory functions (Welte & Gould, 2017). Indeed, although we here highlight a

developing picture of an LD/ROS-dependent trigger of cell death, in other contexts LDs have protective functions against oxidative damage, both in flies and mammals (Bensaad *et al*, 2014; Bailey *et al*, 2015). This may occur by providing an environment that shields fatty acids from peroxidation by ROS and/or by sequestering toxic peroxidated lipids. Although this superficially appears to contradict the theme of LD/ROS toxicity, it is important to recall that in LD-related cell death is not simultaneous with LD accumulation. In fact, degeneration temporally correlates with the dispersal of LDs in middle age, rather than their earlier accumulation. Together, the strands of evidence from several studies suggest that it is the combination of elevated ROS and the dispersal of abnormally high quantities of lipids from previously accumulated LDs that trigger death. This suggests that cells die by being overwhelmed by toxic peroxidated lipids when abnormal accumulations of LDs break down in the presence of high levels of ROS. Our experiments with Brummer lipase are consistent with this idea: the Brummer lipase was expressed from early in development, thereby preventing abnormal LD accumulation, and this protected against cell death. This sequence of events implies the existence of a metabolic switch, when LDs stop accumulating and begin to disperse, triggering the toxic phase of ADAM17 loss. It will be interesting in the future and may provide insights into the normal ageing process, to understand the molecular mechanism of this age- and/or activity-dependent change.

ADAM17 is one of the most important shedding enzymes in humans, responsible for the proteolytic release of a vast array of cell surface signals, receptors and other proteins (Boutet *et al*, 2009; Lorenzen *et al*, 2016; Zunke & Rose-John, 2017). Because of its role in signalling by both TNF and ligands of the EGF receptor, it has been the focus of major pharmaceutical efforts, with a view to treating inflammatory diseases and cancer (Arribas & Esselens, 2009; Mustafi *et al*, 2017). It is therefore surprising that it has been very little studied in *Drosophila*. This is the first report of *Drosophila* ADAM17 mutants. Here, we also confirm for the first time that *Drosophila* ADAM17 is indeed an active metalloprotease, able to shed cell surface proteins including the *Drosophila* TNF homologue Eiger. The only other description of *Drosophila* ADAM17 function is mechanistically consistent with our data, despite relating to a different physiological context. In that case, ADAM17 was shown to cause the release of soluble TNF from the fat body so that it can act as a long range adipokine (Agrawal *et al*, 2016). We cannot rule out other ADAM17 substrates in different developmental or physiological contexts, although the relatively subtle phenotype of null mutant flies implies that ADAM17 does not have essential functions that lead to obvious defects when mutated. Moreover, we have not observed developmental defects or LD accumulation in any neuronal or non-neuronal ADAM17$^{-/-}$ larval tissues, suggesting that the mechanism we report here is both age and tissue specific.

Although TNF is sometimes viewed as a specific cell death-promoting signal, and the pathways by which it activates caspase-induced apoptosis have been studied extensively in flies and mammals (Aggarwal, 2003; Wajant *et al*, 2003) (Wallach *et al*, 2002), the response to TNF is in fact very diverse, depending on the biological context (Postlethwaite & Seyer, 1990; Puls *et al*, 1999). Indeed, its most well-studied role in mammals is as the primary inflammatory cytokine, released by macrophages and other immune cells, and triggering the release of other cytokines, acting as a chemoattractant, stimulating phagocytosis, and promoting other inflammatory responses (Bradley, 2008; Sedger & McDermott, 2014). To our knowledge, however, it has not previously been shown to have trophic activity, protecting cells in the nervous system from stress-induced damage, although a link with the Nrf2/Keap1 redox pathway in cardiomyocytes provides an interesting parallel (Shanmugam *et al*, 2016).

In conclusion, our work highlights three important biological concepts. The first is to identify a new function for the ADAM17/TNF pathway in a cytoprotective role that protects *Drosophila* retinal cells against age- and activity-dependent degeneration. This contrasts with its well-established roles in inflammation and cell death. Secondly, we highlight the existence of a glia-centric cellular pathway by which the breakdown of accumulated LDs and ROS together participate in promoting stress-induced and age-related cell death. Finally, we have shown that the core phenomenon of the ADAM17 protease, acting to regulate the homeostatic relationship between ROS and LD biosynthesis, is conserved in human microglial cells, which themselves are intimately involved in neuroprotection.

# Materials and Methods

### Fly genetics

#### *Generation of ADAM17$^{-/-}$ flies*

ADAM17$^{-/-}$ flies were generated through CRISPR using the guides <u>CTGTGGCCTCGCAATAATCT**CGG**</u> **and** GCC<u>TTTT</u>CGTCGA-GAATTG**CGG** for targeting the first exon of the gene. This caused a 5-bp deletion, resulting in a premature stop codon. We call this mutant allele *ADAM17$^1$*.

These guides were microinjected into $y^1$ *P(act5c-cas9, $w^+$) M (3xP3-RFP.attP)ZH-2A $w^*$* (Cambridge microinjection facility) and then subsequently screened using Melting curve analysis.

#### *Transgenic flies*

To generate the UAS-ADAM17 WT fly lines, the *pUASattB-ADAM17 WT* construct was introduced into the germ line by injections in the presence of the PhiC31 integrase and inserted in the attp40 landing site on the 2nd chromosome (Cambridge microinjection facility).

The following lines were from the GD or KK collections of the Vienna *Drosophila* RNAi Centre (VDRC): *ADAM17* RNAi (v2733), *grnd* RNAi (43454), *wengen* RNAi (58994), *kuz* RNAi (v107036).

The *$w^{1118}$, elav-Gal4 (8765)*, *actin-GAL4*(BL25374), *sparkling-GAL4* (BL 26656), *rh1-GAL4* (BL 68385), *gmr-GAL4* (BL9146), *54C-GAL4* (27328), *eiger* RNAi(58993), *ADAM17 Deficiency* (27366), *UAS-bmm* (76600), *UAS-lacZ* (1776), *$bsk^1$* (3088), *UAS-SOD2* (24494), *UAS -SOD1* (24491) and *UAS-catalase* (24621) were obtained from the Bloomington Stock centre. The mutant alleles for Grindelwald and Wengen were obtained from Bloomington : *$grnd^{Minos}$* CG10176MI05292, BL43677 and BL17874, respectively. The mutant allele for eiger used in this study is *$egr^1$* (Igaki *et al*, 2002).

All crosses were performed at 29°C in light, unless otherwise stated.

### Fly genotypes
**Fig 1**

$w^{1118}$;
+/+;$ADAM17^{-/-}$;
+/+;$ADAM17^-/Df$
;GMR-GAL4/+; UAS-lacZ/+
;GMR-GAL4/+; UAS-ADAM17$_i$/+
;elav-GAL4/+; UAS-lacZ/+
;elav-GAL4/+; UAS-ADAM17$_i$/+
spa-GAL4;+/+;UAS-lacZ/+
spa-GAL4;+/+; UAS-ADAM17$_i$/+
;$kuz^{-/-}$

**Fig 2**

$w^{1118}$
;+/+;$ADAM17^{-/-}$
;+/+;$ADAM17^-/Df$
;+/+;Actin-GAL4/UAS-lacZ
;+/+;Actin-GAL4/UAS-ADAM17$_i$
spa-GAL4;+/+;UAS-lacZ/+
spa-GAL4;+/+;UAS-ADAM17$_i$/+
;elav-GAL4/+;UAS-lacZ/+
;elav-GAL4/+;UAS-ADAM17$_i$/+
;$kuz^{-/-}$

**Fig 3**

$w^{1118}$
;+/+;$ADAM17^{-/-}$
spa-GAL4;UAS-bmm/+;
spa-GAL4;UAS-bmm/+;$ADAM17^{-/-}$
,elav-GAL4/UAS-bmm;$ADAM17^{-/-}$

**Fig 4**

$w^{1118}$
;$egr^{-/-}$
;$grnd^{-/-}$
$wgn^{-/-}$

**Fig 5**

$w^{1118}$
;+/+;$ADAM17^{-/-}$
spa-GAL4;+/+;$ADAM17^{-/-}$
spa-GAL4; UAS-SOD2/+;$ADAM17^{-/-}$
spa-GAL4; UAS-SOD1/+;$ADAM17^{-/-}$
;elav-GAL4;UAS-SOD2/+; $ADAM17^{-/-}$

**Fig 6**

$w^{1118}$
;+/+;$ADAM17^{-/-}$
;$bsk^-/+$

;$bsk^-/+$; $ADAM17^{-/-}$

**Fig EV1**

;+/+;Rh1-GAL4/UAS-lacZ/+
;+/+;Rh1-GAL4/ UAS-ADAM17$_i$/+ spa-GAL4;+/+;$ADAM17^{-/-}$
spa-GAL4;UAS-ADAM17-WT/+;$ADAM17^{-/-}$
spa-GAL4;UAS-ADAM17-WT/+;
$w^{1118}$
;;$ADAM17^{-/-}$

**Fig EV2**

$w^{1118}$
;+/+;$ADAM17^{-/-}$

**Fig EV3**

$w^{1118}$
;+/+;$ADAM17^{-/-}$

**Fig EV5**

$w^{1118}$
spa-GAL4;;$ADAM17^{-/-}$
spa-GAL4;UAS-SOD2/+;$ADAM17^{-/-}$
spa-GAL4;UAS-SOD1/+;$ADAM17^{-/-}$
spa-GAL4;UAS-catalase/+;$ADAM17^{-/-}$

### Electron microscopy

Samples were fixed in 2.5% Glutaraldehyde + 4% PFA + 0.1% tannic acid in 0.1 M PIPES pH 7.2, for 1 h at room temperature (RT) and then overnight at 4°C. Samples were then washed with 0.1 M PIPES at RT over the course of 2 h with several solution changes, including one wash with 50 mM glycine in 0.1 M PIPES to quench free aldehydes. Samples were then incubated in 2% osmium tetroxide + 1.5% potassium ferrocyanide in 0.1 M PIPES for 1 h at 4°C with rotation. Post-fixation, samples were washed three times with MQ water for 10 min each. This was followed by tertiary fixation with 0.5% uranyl actetate (aq.) at 4°C overnight. All subsequent steps were performed on a rotator set to medium-high speed. Samples were washed three times with MQ water and then sequentially dehydrated for 10 min each in ice-cold 30, 50, 70, 100% ethanol twice, anhydrous ice-cold acetone and finally in anhydrous acetone at RT. Samples were then infiltrated at RT with Durcupan resin as follows: 25% resin in acetone for 3 hrs, 50% resin in acetone overnight, 75% resin for 2–3 h, 100% resin for ~ 6 hrs, 100% resin overnight, 100% resin 7 h, 100% resin overnight and 100% resin 4–5 h. After each change into fresh resin, the samples were spun for 30 s in a minifuge at 4,000 $g$ to aid infiltration. Samples were then embedded in the fresh resin in flat moulds and cured at 60°C for 48 h.

Eyes were sectioned tangentially using a Leica UC7 ultramicrotome. Ultrathin (90 nm) sections were obtained using a Diatome diamond knife and transferred to formvar-coated copper or copper/

palladium 2 × 1 mm slot grids. Sections were post-stained for 5 min with Reynold's lead citrate and washed three times with warm water, blotted and air dried.

Grids were imaged on either a FEI Tecnai 12 Transmission Electron Microscope (TEM) at 120 kV using a Gatan OneView camera or, for low magnification mapping, on a Zeiss Sigma 200 Field Emission Gun Scanning Electron Microscope (FEG-SEM) at 20 kV using the STEM detector.

Fixation and dehydration for Scanning Electron Microscopy (SEM) were carried out similarly to those for the TEM. After dehydration through an ascending series of ethanol concentrations ending in 100% ethanol, samples were dried using critical point drying (CPD), without introducing surface tension artefacts. After drying, the samples were carefully mounted on an aluminium stub using silver paint. Samples were then introduced into the chamber of the sputter coater and coated with a very thin film of gold before SEM examination. All SEM samples were imaged on the Zeiss Sigma 200 Field Emission Gun Scanning Electron Microscope (FEG-SEM) at 20 kV.

## Degeneration measurements

Missing, abnormal or missing rhabdomeres were counted manually on TEM images from a total of 60 ommatidia per retina using ImageJ. These counts were then averaged over three retinas per genotype. The average percentage for each genotype is displayed as a stacked column graph.

## Whole-mount staining of *Drosophila* retinas and imaginal discs

For LD staining, *Drosophila* adult heads were cut in half and the brain was removed to expose the retina underneath. Larval imaginal discs and pupal retinas were dissected in ice-cold PBS. Tissues were fixed in 3.7% formaldehyde (Sigma, F8775-500ML) in PBS for 15 min and washed three times in PBS1X. Tissues were incubated for 10 min in BODIPY™ 558/568 (Invitrogen, D3835) to label lipid droplets, and FM™ 4-64FX (Invitrogen, F34653) to stain membranes. Retinas were then mounted in Vectashield (Vector Laboratories Ltd, H-1000) and imaged on Olympus FV1000.

## Lipid droplet measurements

Lipid droplet numbers and size were measured from the BODIPY signal, using an algorithm developed on ImageJ. Integrated total lipid was calculated as a product of normalised size and total lipid droplet numbers across an individual retina. Normalised LD size was calculated by dividing the size of LDs across all genotypes with that of its corresponding control.

## Immunostaining of whole-mount *Drosophila* retinas

For immunostaining, retinas were dissected similarly as above. Retinas were fixed in 4% paraformaldehyde (VWR, 43368.9M) in PBS for 15 min and washed three times in PBS1X-triton 0.3%. Retinas were labelled with anti-ADAM17 (Abcam) overnight at 4°C followed by secondary antibody and phalloidin staining (Life Technologies, A12380) for 2 h at room temperature. Retinas were mounted in Vectashield (Vector Laboratories Ltd, H-1000) and imaged on Olympus FV1000.

## Line intensity profiles

Line intensity profiles for ADAM17 and phalloidin were calculated manually using the Plot profile plugin on ImageJ. Intensity values were measured along a line in FIJI averaging 10 m, that spanned a single ommatidia, and plotted along its distance.

## mRNA extraction and quantitative PCR

Twenty frozen fly heads were used to extract RNA using TRIzol reagent with the help of a Direct- zol mRNA kit, according to the manufacturer's instructions. cDNA was prepared from 0.4 μg RNA using the qpcrbiosystems kit (PB30.11-10). Resulting cDNA was used for Quantitative PCR (qPCR) combining it with TaqMan Gene Expression Master Mix (Applied Biosystems) and TaqMan probes (all Thermo Fisher). qPCR was performed on a StepOnePlus system Thermocycler (Applied Biosystems). Each biological experiment was carried out with three independent technical replicates and normalised to actin in each case. Error bars indicate the standard error from mean for at least three different biological replicates. TaqMan probes used for all experiments are as follows: *Drosophila* ACC (Dm01811991_m1), *Drosophila* FASN (Dm01821412_m1), *Drosophila* TACE/ADAM17(Dm02146367_g1), *Drosophila* actin (Dm02362162_s1), Human FASN (Hs01005624), Human LDLR (Hs00181192_m1) and Human Actin (Hs01060665_g1).

## Statistical analysis for LD and Q-PCR measurements

All datasets were analysed in Prism. Normality and homogeneity of variance were used to determine whether the data met the assumptions of the statistical test used. All datasets were assumed to be independent. Datasets with unequal variance were analysed using the Kruskal–Wallis test followed by Dunn's test for *post hoc* analysis for significance due to unequal sample sizes. All other datasets were quantified for significance using Student's *t* test. Significance is defined as *P < 0.05, **P < 0.01 and ***P < 0.001, and error bars are shown as standard error of the mean (SEM) unless otherwise noted. For fly experiments, more than 10 flies were used for each individual experiment, and all crosses were performed at least twice. For cell experiments, all studies were conducted in parallel with vehicle controls in the neighbouring well, for at least three wells (biological replicates). All data points, including outliers, were used for statistical analysis.

## Western blotting

Ten frozen fly head heads or 20 adult retinas were lysed in 50 μl ice-cold buffer (20 mM Tris-HCl, 100 mM NaCl, 1% IGEPAL and 2 mM in water) supplemented with Protease Inhibitors (Sigma, MSSAFE-5VL) and Benzonase (Sigma, E1014-24KU) for 25–30 min on ice. These samples were centrifuged at 20,000 *g* for 45 min at 4°C. The supernatant from each tube was then mixed with sample buffer (NuPAGE), supplemented with DTT and incubated at 65°C for 15 min. For Oxyblot measurements, supernatants were sequentially treated with 12% SDS, 1× 2-2-dinitrophenyl hydrazine (DNPH) and 1× neutralisation solutions, according to the manufacturer's instructions (Sigma, S7150), before proceeding to SDS–PAGE analysis.

Proteins were resolved by SDS–PAGE using 4–12% Bis-Tris gels (NuPAGE, Life Technologies) and transferred via electrophoresis to polyvinylidene difluoride membranes (PVDF, Millipore). Membranes were blocked for 1 h at room temperature in blocking buffer (5% milk in 0.1% triton in PBS/PBST) and incubated overnight at 4°C in the same buffer containing primary antibodies at 1:1,000 dilution. Membranes were washed three times in PBST and then incubated with secondary antibodies in milk, for 1 h. Post this step, membranes were again washed three times in PBST and then developed using ECL reagents (Thermo Scientific).

## Cloning of murine and *Drosophila* ADAM17 constructs and stably expressing cell lines

Murine ADAM17 (mA17), *Drosophila* ADAM17 (dA17) and its mutants each with a PC tag were cloned into the pMOWS vector for retrovirus-based transduction (Ketteler *et al*, 2002). For the inactive HE mutant of *Drosophila* ADAM17 (dADAM17), the following single point mutations were introduced: E400H, H409E. HEK293 cells deficient for ADAM17 and ADAM10 (HEK293_A17$^{-}$_A10$^{-}$) (riethmueller) stably expressing the different ADAM17 constructs were generated via the retroviral-based pMOWS/Phoenix ampho system. 50 μg/ml Zeocin was used as a selection marker.

## Shedding activity assay in HEK cells

Shedding activity of ADAM17 variants was measured by an alkaline phosphatase (AP)-based assay. For this assay, $5 \times 10^6$ cells were seeded on a 10-cm dish and transiently transfected with the ADAM17 substrate IL-1R$_{II}$ fused to an alkaline phosphatase (AP). Transient transfection was performed via the use of Lipofectamine 3000 (Thermo Fischer Scientific). After 24 h, $2 \times 10^5$ cells/well were transferred into 24-well plates. 24 h later, cells were treated with only 100 nM PMA, 100 nM PMA combined with 10 μM TAPI-1 or treated with the corresponding volume of vehicle (DMSO). Cells were incubated for 120 min at 37°C. The activity of AP in cell lysates and supernatants was determined by incubating 100 μl AP substrate p-nitrophenyl phosphate (PNPP) (Thermo Scientific) with 100 μl cell lysate or cell supernatant at room temperature followed by the measurement of the absorption at 405 nm. The percentage of AP-conjugated material released from each well was calculated by dividing the signal from the supernatant by the sum of the signal from lysate and supernatant. The data were expressed as mean of at least three independent experiments. All data points, including outliers, were used for statistical analysis.

## Statistical analysis of AP assay in HEK cells

Quantitative data are shown as mean with standard deviation (SD) calculated from $n = 10$ independent experiments. Statistics were conducted using the general mixed model analysis (PROC GLIMMIX, SAS 9.4, SAS Institute Inc., Cary, NC, USA) and assumed to be from a lognormal distribution with the day of experiment conduction as random, to assess differences in the size of treatment effects across the results. Residual analysis and the Shapiro–Wilk test were used as diagnostics. In the case of heteroscedasticity (according to the covtest statement), the degrees of freedom were adjusted by the Kenward–Roger approximation. All *P*-values were adjusted for multiple comparisons by the false discovery rate (FDR). $P < 0.05$ was considered significant. All data points, including outliers, were used for statistical analysis.

## Activity assay in S2R+ cells

Shedding activity of ADAM17 variants was measured by an alkaline phosphatase (AP)-based assay. 24-well plates were first coated with Poly-L-lysine (Sigma). Around $4 \times 10^5$ cells/well were plated onto 24-well plates containing a mix of ADAM17 variants with eiger fused to AP plasmids and transfection reagent (Fugene). Media was changed after 24 h. 48 h, post-plating cells were treated with only 100 nM PMA, 100 nM PMA combined with 10 μM TAPI-1 or treated with the corresponding volume of vehicle (DMSO). Cells were incubated for 120 min at 37°C. Cell supernatants were collected, and the cells were washed in PBS and lysed in 200 μl Triton X-100 lysis buffer. The activity of AP in cell lysates and supernatants was determined by incubating 100 μl AP substrate p-nitrophenyl phosphate (PNPP) (Thermo Scientific) with 100 μl cell lysate or cell supernatant at room temperature followed by the measurement of the absorption at 405 nm. The percentage of AP-conjugated material released from each well was calculated by dividing the signal from the supernatant by the sum of the signal from lysate and supernatant. The data were expressed as mean of at least three independent experiments.

## MDA analysis of peroxidated lipids

The Lipid Peroxidation (MDA) Assay Kit (Colorimetric/Fluorometric) (Abcam, Cat# ab118970) was used as per the manufacturer's instructions. 1 million cells or fifteen retinas per sample were dissected in cold 1× PBS and then transferred to 120 μl MDA lysis buffer with 1 μl BHT. After homogenisation, samples were vortexed and centrifuged to remove precipitated protein. 100 μl of the supernatant was added to 300 μl thiobarbituric acid reagent and incubated at 95°C for 1 h. 200 μl of standard or sample was added to individual wells of a GREINER 96 F-BOTTOM plate. For fluorometric measurement, signals were collected with a CLARIOStar reader (BMG LABTECH GmbH) (Ex/Em = 532 ± 8/553 ± 8 nm). Four biological replicates were quantified per sample. All data points, including outliers, were used for statistical analysis.

## Human induced Pluripotent stem cells

The iPS cell lines used in this study have all been published previously (SFC840-03-03(Fernandes *et al*, 2016), SFC841-03-01(Dafinca *et al*, 2016), SFC856-03-04(Haenseler *et al*, 2017) and are all available from the European Bank for iPS cells, EBiSC. They were derived in the James Martin Stem Cell Facility, University of Oxford, from dermal fibroblasts of healthy donors, who had given signed informed consent for the derivation of iPSC lines from skin biopsies as part of the Oxford Parkinson's Disease Centre (Ethics Committee: National Health Service, Health Research Authority, NRES Committee South Central, Berkshire, UK (REC 10/H0505/71). All experiments used cells thawed from large-scale karyotype QCed frozen stocks.

iPS cells were differentiated through a primitive myeloid differentiation pathway to generate primitive macrophage precursors (Buchrieser *et al*, 2017), followed by maturation according to (Haenseler *et al*, 2017) to microglial-like cells, seeding at 75,000 cells per well on 96 Well Black-walled imaging plates (Ibidi 89626) precoated for 1 h with geltrex (Life Technologies A1413302), and fed twice-weekly for 2 weeks with 150 µl microglial medium. Microglial medium was generated to have a final concentration of 1× Advanced DMEM/F12 (Life Technologies, 12634-010), 1× N2 supplement (Life Technologies, 17502-048), 2 mM GlutaMAX (Life Technologies, 35050-061), 50 µM 2-mercaptoethanol (Life Technologies, 31350-010), 50 U/ml Pen/Strep (Life Technologies, 17502-048), 100 ng/ml IL-34 (Peprotech, 200-34) and 10 ng/ml GM-CSF (LifeTechnologies, PHC2013).

### Staining and imaging of human iPS cells

Drug treatments of either 3 µm GW (synthesised by Iris Biotech), 3 µm GI (synthesised by Iris Biotech), 10 µm TAPI-1 (Cayman Chemicals) or DMSO (Sigma) were performed by supplementing the media with the above drugs for cells grown on 96 well ibdi Plates, suitable for imaging. Cells were then incubated with either 5 nM MitoSOX (Thermo Fischer Scientific) for 30 min at 37°C, 200 nM C-11 BODIPY (Thermo Fisher Scientific) for 30 min at 37°C or 493/503 BODIPY (Thermo Fisher Scientific) and FM 4-64FX (Thermo Fisher Scientific) dyes, for 10 min at 37°C. Post-staining cells were almost immediately imaged on the Live Cell Olympus FV1200, maintaining the temperature at 37°C throughout the course of the experiment.

### Lipid droplet measurements on iPS cells

Lipid droplet numbers and size were measured using an algorithm developed on ImageJ. Integrated total lipid was calculated as a product of normalised size and total lipid droplet numbers across an individual cell. Normalised LD size was calculated by dividing the size of LDs across all genotypes with that of its corresponding control.

### Image analysis and compilation

Post-acquisition, all images were analysed on ImageJ and subsequently compiled on Adobe Photoshop. The model was compiled on Adobe Illustrator.

## Data availability

This study contains no data deposited in external repositories.

Expanded View for this article is available online.

### Acknowledgements

We thank Bertrand Mollereau, Viorica L Lastun, Alessia Gelasso, Chaitali Khan and Mike Renne for their insightful comments on the manuscript. We also thank Mike Murphy, Pedro Carvalho and all Freeman Lab members for useful discussions; Antonio Baonza for help with pupal dissections; Ni Tang for help with development of the lipid droplet quantifying algorithm; and Pierre Leopold for sharing fly stocks. We particularly thank Errin Johnson and members of the Dunn school EM facility for advice on EM protocols. All confocal experiments were performed at the Micron Oxford Advanced Bioimaging Unit. SM was funded by HFSP and a non-stipendiary EMBO fellowship. SD was funded by a Research Fellowship of the German Research Foundation (DU 1582/1-1). The James Martin Stem Cell Facility (SAC) has received financial support from the MRC (MC_EX_MR/N50192X/1) and the Oxford Parkinson's Disease Centre Monument Trust Discovery Award from Parkinson's UK (grant J-1403). This work was primarily supported by a Wellcome Trust Investigator Award (101035/Z/13/Z) to MF.

## Author contributions

SM and MF conceptualised and wrote the original draft of the paper. SM, CL and MF designed the methodology for experiments. SM and CL conducted all fly-related experiments. SM did all the analysis and iPSC cell culture experiments. SD and ID performed cloning of constructs and AP-shedding in HEK cells. SAC provided iPS cells and advised on their experimental use.

## Conflict of interest

The authors declare that they have no conflict of interest.

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
