## [Review Process File · The EMBO Journal]

ADAM17-triggered TNF signalling protects the ageing *Drosophila* retina from lipid droplet mediated degeneration

Sonia Muliyl, Clemence Levet, Stefan Düsterhöft, Iqbal Dullo, Sally Cowley, and Matthew Freeman
DOI: [10.15252/embj.2020104415](https://doi.org/10.15252/embj.2020104415)

Corresponding author(s): Matthew Freeman (matthew.freeman@path.ox.ac.uk)

Review Timeline:

Submission Date:	7th Jan 20
Editorial Decision:	21st Feb 20
Revision Received:	21st Apr 20
Editorial Decision:	29th May 20
Revision Received:	26th Jun 20
Accepted:	29th Jun 20

Editor: Karin Dumstrei

Transaction Report:

Dear Matthew,

Thank you for submitting your manuscript to The EMBO Journal. I am sorry for the slight delay in getting back to you with a decision, but I have now received the two referee reports on the manuscript.

As you can see below, the referees appreciate the analysis and are supportive of publication here. They raise a number of good suggestions for how to further strengthen the findings. The experiments should all be doable to do within a reasonable timeframe.

Let me know if we need to discuss any of the revisions in more detail as I am happy to do so.

When preparing your letter of response to the referees' comments, please bear in mind that this will form part of the Review Process File, and will therefore be available online to the community. For more details on our Transparent Editorial Process, please visit our website:

<https://www.embopress.org/page/journal/14602075/authorguide#transparentprocess>

Thank you for the opportunity to consider your work for publication. I look forward to your revision.

with best wishes

Karin

Karin Dumstrei, PhD
Senior Editor
The EMBO Journal

- a point-by-point response to the referees' comments, with a detailed description of the changes made (as a word file).

- a word file of the manuscript text.

- individual production quality figure files (one file per figure)

- a complete author checklist, which you can download from our author guidelines

(<https://www.embopress.org/page/journal/14602075/authorguide>).

- Expanded View files (replacing Supplementary Information)

Further information is available in our Guide For Authors:

The revision must be submitted online within 90 days; please click on the link below to submit the revision online before 21st May 2020.

Link Not Available

Referee #1:

The authors generate ADAM17 null mutant flies that accumulate lipid droplets (LDs) in the Pigmented Glial Cells (PGCs) of the eye immediately after eclosion. LDs subsequently decrease, accompanied by progressive degeneration of PGC and photoreceptor cells over the following weeks. Knock down or overexpression of ADAM17 in PGCs is sufficient to phenocopy or rescue the mutant, respectively. Increased LD abundance in the adult eye correlates with PGC-specific upregulation of lipogenic genes ACC and FASN. Overexpression of Bmm/ATGL lipase in the PGCs prevents LD accumulation and subsequent degeneration in ADAM17 mutants, suggesting that LD accumulation is detrimental in this system. ADAM17 cleaves the TNF ligand Eiger in S2R cells and Eiger and the receptor Grindelwald act in PGCs to trigger ACC and FASN upregulation, LD accumulation and retinal degeneration. Returning to the ADAM17 mutant phenotype, they find that this is rescued by AD4 antioxidant or PGC>SOD2 overexpression, or by keeping the flies in the dark to block photoreceptor activity. This suggests that ADAM17 suppresses ROS, consistent with the increased MDA in ADAM17^{-/-} heads. JNK activity is also required for the LD and degeneration phenotypes, placing JNK downstream of ADAM17. Finally, they use human iPSC derived microglia to show that chemical inhibition of ADAM17, but not ADAM10, is sufficient to trigger similar responses to those seen in the fly eye: increased lipogenic gene transcription, LDs, mitochondrial ROS and lipid peroxidation.

This clearly written paper adds to a growing literature surrounding the origins and roles of LDs during retinal degeneration. The experiments are generally well controlled and presented, and the data largely support the newer published interpretations of LD functions in the fly retinal glia (Liu 2017, Van Den Brink 2018) - namely that PGC lipid droplets are not detrimental per se but enhancing lipid droplet lipolysis (via Bmm/ATGL) is protective under high ROS conditions. The key and interesting new finding here is the antioxidant role of ADAM17 in the retina, acting via the *Drosophila* TNF/TNFR homologues Eiger and Grindelwald in the PGCs.

MAJOR:

1. "The ADAM17 mutants we report are different, showing no evidence for abnormal lipid production in neurons" (p13). To support that their conclusions are different (from the Bellen lab papers) and also to support their Figure 8 model, the authors should demonstrate (perhaps using published fly retina methods to knockdown lipoprotein carriers in the two different neurodegeneration models) that abnormal lipids derived from the photoreceptor neurons of ADAM17 mutants are not driving lipid droplet accumulation in PGCs. This is important given the Bellen and Lippincott-Schwartz lab papers found the opposite, and one of these studies (Ioannou et al, 2019) related this to neuronal hyperactivity - perhaps paralleling the light requirement in this manuscript, Muliyl et al.

2. As newly eclosed ADAM17 mutant flies have abnormally enlarged PGCs (page 5), the authors cannot rule out a developmental role (e.g during pupal development) rather than a purely adult degeneration role. Perhaps the most important damage is already done in GMR-GAL4 expressing PGCs before adult eclosion and this then triggers a progressive photoreceptor decline through adulthood. In other words, could it be that the critical period for TNF signalling in PGCs is during pupation and not during adult aging? One possible way to resolve this would be with GAL80[ts] temperature shift experiments. Related to this, the authors should show whether ADAM17 is or is not expressed in the pupal retina.

3. The use of the dye BODIPY 493/503 is a good marker for neutral lipids. But are the authors sure the neutral lipids they detect are always inside lipid droplets, or could it be also in other organelles such as lysosomes (via lipophagy)? Use of lipid droplet markers (such as Lsd2::GFP) and perhaps lysosome markers would help to distinguish this.

4. How does TNF regulate ROS? "To our knowledge, however, it has not previously been shown to have trophic activity, protecting cells from stress-induced damage, in the nervous system or elsewhere" (page 15). A few minutes Googling revealed at least one paper proposing a protective function for basal TNF- α signaling in cardiomyocytes, where it may act upstream of the well-known antioxidant Keap1/Nrf2 pathway (<https://www.ncbi.nlm.nih.gov/pmc/articles/PMC4961303/>). Does this pathway lie downstream of ADAM17/TNF in the PGC?

MINOR:

1. PGC LDs are known to be downstream of ROS-induced JNK, which may activate SREBP to drive lipogenesis (Liu et al, 2015). It would seem likely that a similar mechanism accounts for the LD changes reported in this paper. Data should be presented on this or the JNK - SREBP connection at least discussed.

2. In the abstract, it is stated that the 'damage is caused by neuronally generated ROS' whereas only modest rescue of the degenerative phenotype occurs when neuronal ROS are prevented. The

authors need to remove the term 'neuronally' as they provide evidence that it is mainly glial.

3. page 7. Incorrect figure citation for Figure S4B. Should this be Figure S5B?

4. page 8. Overinterpretation: "...indicating that the proteolytic activity of ADAM17 is indeed necessary...". TAPI-1 containing food will surely inhibit more proteases than just ADAM-17.

5. page 13. The two citations of Fig 4G in the Discussion appear to be out of context and do not to make sense. Please clarify.

6. The Figure 8 model refers to neurons and glia, not photoreceptors and PGCs. This expansion of the paradigm from the eye to the generic nervous system is not justified from the data presented.

NON-ESSENTIAL:

1. Strengthening the ADAM17 - TNF connection in the in vivo context of retinal degeneration would be valuable, since this is the major new finding to the existing PGC-photoreceptor degeneration models. A key prediction is that PGC expression (at or near endogenous levels) of cleaved Eiger or activated Grindelwald will rescue the ADAM17^{-/-} phenotype.

Referee #2:

In this study, Muliyl and colleagues report that the TNF signaling cascade in *Drosophila*, mediated by ADAM17, Eiger and Grnd, protects pigment glial cells (PGCs) of the eye from degeneration. They specifically show that ADAM17 mutants exhibit a progressive retinal degeneration phenotype, and show that this reflects ADAM17's requirement in the PGCs. The authors report a similar phenotype with loss of function mutants of Eiger (*Drosophila* TNF α family ligand) and Grnd (TNF receptor). Evidence presented here indicates that in ADAM17 mutants, the PGCs accumulate Reactive Oxygen Species (ROS) in the mitochondria, which causes the activation of JNK signaling to induce lipid biogenesis for lipid droplet (LD) formation. Breaking down LDs early on suppresses the degeneration phenotype. However, the onset of retinal degeneration correlates with the dispersal of LD droplets, indicating that there are two temporal phases of LD formation that affects PGC degeneration in distinct ways. The authors use iPSC cells to demonstrate that ADAM17 has similar roles in mammalian cultured cells.

Overall, this is a very nicely executed study that establishes a previously unrecognized role of ADAM17 in the protection of cells from ROS and LD-mediated neurodegeneration. The quality of the data is very high, with proper controls and validation of critical observations through multiple independent means. I only have a few minor suggestions that could further enhance the clarity of the manuscript.

1. The authors used two guide RNAs (gRNA) to generate an ADAM17 mutant allele. The methods section describing this mutant is rather brief, mentioning that the mutants were screened using a "melting curve analysis." The authors should explain whether the mutant line that they used has an actual deletion between the two gRNA targeted sites or not. Also, it would be useful to give the mutant allele a name.
2. The authors do not describe the source of Eiger, Grnd, and wgn loss-of-function mutant alleles used in Figure 4 (also Figure S6) in the Methods section. Which alleles did they use?
3. In Figure 4L - P, the authors show the effect of Eiger, grnd, and wgn loss on neuronal

degeneration at two weeks of age. Why was this time point chosen when their analyses of adam17 loss-of-function were primarily done at five weeks of age (e.g., Figures 1 and 3)? The degeneration of Eiger mutants at two weeks of age, as shown in Figure 4, is somewhat modest. I wonder if the authors could report a more clear difference if they were to examine five-week old flies.

EMBOJ-2020-104415. Muliyl et al. Responses to reviewers

Reviewer 1

MAJOR:

1. *"The ADAM17 mutants we report are different, showing no evidence for abnormal lipid production in neurons" (p13). To support that their conclusions are different (from the Bellen lab papers) and also to support their Figure 8 model, the authors should demonstrate (perhaps using published fly retina methods to knockdown lipoprotein carriers in the two different neurodegeneration models) that abnormal lipids derived from the photoreceptor neurons of ADAM17 mutants are not driving lipid droplet accumulation in PGCs. This is important given the Bellen and Lippincott-Schwartz lab papers found the opposite, and one of these studies (Ioannou et al, 2019) related this to neuronal hyperactivity - perhaps paralleling the light requirement this manuscript, Muliyl et al.*

Response: This is an important point that we tried to highlight in our manuscript because it does indeed distinguish our results from those reported by the Bellen and Lippincott-Schwartz groups. We emphasise that we do not think the data are in any way incompatible, just that the conditions that lead to stress are different in our experiments. In their cases they stressed the retina by knocking down essential mitochondrial genes in neurons; in ours, it is the loss of ADAM17 from glia that causes the accumulation of LDs and degeneration.

All our data make the point strongly that ADAM17 is required only in glia, not photoreceptor neurons, to protect against retinal degeneration. Reviewer 1 does not question this. The more specific question is the source of the lipids that contribute the glial LD accumulation: might they be made in adjacent neurons and transported to the glia? Our best evidence against this is that expression of the lipase Brummer in glia prevents the accumulation of LDs and neurodegeneration; in contrast, expression of lipase in neurons had no measurable effect on LD accumulation. This implies that the excess lipids are not substantially being generated by neurons. Significantly, it contrasts with the results for lipase overexpression in similar experiments by the Bellen group, again highlighting the differences between the two contexts. Nevertheless, when we looked at the less direct measure of retinal degeneration, we did see a low level of rescue when Brummer was expressed in neurons. Since we saw no effect on LD accumulation itself, it is hard to interpret this as a clear sign of neuronal involvement as a source of accumulating lipids, but we acknowledge that we cannot rule out a minor contribution.

Overall, we remain confident that our data strongly support the conclusion that glia are the primary source of excess lipids, but we agree that we cannot rule out some contribution of neurons, so we have adjusted the text to make this explicit. The experiment proposed by Reviewer 1, to knock down the different lipid transporters in neurons, is a good idea in principle. However, with our lab closed because of the coronavirus pandemic for the foreseeable future, and it then requiring us to obtain stocks from other potentially closed labs, followed by several fly generations to make the needed stocks, we suspect this would end up taking a minimum of 6 months. We hope that in the circumstances, our proposed approach will be seen as fair and proportionate.

2. *As newly eclosed ADAM17 mutant flies have abnormally enlarged PGCs (page 5), the authors cannot rule out a developmental role (e.g during pupal development)*

rather than a purely adult degeneration role. Perhaps the most important damage is already done in GMR-GAL4 expressing PGCs before adult eclosion and this then triggers a progressive photoreceptor decline through adulthood. In other words, could it be that the critical period for TNF signalling in PGCs is during pupation and not during adult aging? One possible way to resolve this would be with GAL80[ts] temperature shift experiments. Related to this, the authors should show whether ADAM17 is or is not expressed in the pupal retina.

Although the results that we had already included indicated progressive degeneration through adult life, with no obvious damage at eclosion, we agree that it is possible that the initial damage caused by loss of ADAM17 occurs in pupal life. We tried but were unable to perform the suggested Gal80[ts] experiments because the necessary heatshocks themselves induced stress and LD accumulation, making analysis impossible. We have however addressed this question in two other ways.

1. We analysed LD numbers in 40 hour pupae of wild type and ADAM17 mutants. We detected no significant difference in LD numbers or morphology, implying that at least by that stage (the latest when it is practical to dissect pupal retinas), loss of ADAM17 has no detectable phenotype.
2. We also looked at the expression of ADAM17 in pupal retinas both at the RNA and protein level. In both cases, ADAM17 is expressed only at low levels in 40 hour pupal retinas, compared to the stronger expression seen in glial cells at eclosion.

These results reinforce the conclusion that the critical period for ADAM17 function is from eclosion [Figure EV3]. We accept, however, that it is not possible to rule out a minor earlier effect, particularly in the latest pupal stages, just prior to eclosion. We have therefore added this point to the main text.

3. The use of the dye BODIPY 493/503 is a good marker for neutral lipids. But are the authors sure the neutral lipids they detect are always inside lipid droplets, or could it be also in other organelles such as lysosomes (via lipophagy)? Use of lipid droplet markers (such as Lsd2::GFP) and perhaps lysosome markers would help to distinguish this.

We have addressed this question in two ways.

1. We have now performed a co-immunofluorescence experiment with both BODIPY and lysotracker to view respectively LDs and lysosomes in wild type and ADAM17 mutant adult retinas. There was no colocalisation the two markers, as measured using the Coloc2 Plugin in ImageJ (Pearson's correlation coefficient <0.02). This indicates that the vast majority of lipid-rich puncta are not lysosomal.
2. We have also analysed by western blot the levels of the LSD2 protein, which is a LD surface marker, in head lysates of wild type, ADAM17 and eiger mutants. This shows a clear increase in LD markers in the genotypes with excess lipid puncta staining. Again, this argues that the puncta we see are indeed bona fide LDs.

We have added these results to Figure S1.

4. How does TNF regulate ROS? "To our knowledge, however, it has not previously been shown to have trophic activity, protecting cells from stress-induced damage, in the nervous system or elsewhere" (page 15). A few minutes Googling revealed at least one paper proposing a protective function for basal TNF- α signaling in cardiomyocytes, where it may act upstream of the well-known antioxidant Keap1/Nrf2 pathway (<https://www.ncbi.nlm.nih.gov/pmc/articles/PMC4961303/>). Does this pathway lie downstream of ADAM17/TNF in the PGC?

We thank Reviewer 1 for pointing out that paper, which we had missed. We have now analysed the expression of some of the targets of the Nrf-Keap1 pathway discussed in that paper (catalase, SOD2 and gclc). Interestingly, these targets show reduced expression in ADAM17 mutants, suggesting that a weakened antioxidant response may contribute to the effects we report, and that the Nrf2-Keap1 pathway might lie downstream of ADAM17/TNF signalling. We have added this point to Figure EV5 and have referred to it in the Discussion.

MINOR:

1. PGC LDs are known to be downstream of ROS-induced JNK, which may activate SREBP to drive lipogenesis (Liu et al, 2015). It would seem likely that a similar mechanism accounts for the LD changes reported in this paper. Data should be presented on this or the JNK - SREBP connection at least discussed.

We tried an SREBP antibody, but it was not conclusive, owing to a lack of a fly specific SREBP antibody. Instead, we have now incorporated a point in the discussion about the JNK-SREBP connection.

2. In the abstract, it is stated that the 'damage is caused by neuronally generated ROS' whereas only modest rescue of the degenerative phenotype occurs when neuronal ROS are prevented. The authors need to remove the term 'neuronally' as they provide evidence that it is mainly glial.

We believe that majority of the ROS is neuronally generated. This is because in the absence of light, which is expected to lower specifically neuronal ROS, the degeneration in ADAM17 mutants is substantially reduced. But we have amended the text of the abstract to make the claim less sweeping.

3. page 7. Incorrect figure citation for Figure S4B. Should this be Figure S5B?

Thanks for pointing us towards the incorrectly cited figure. This has now been corrected.

4. page 8. Overinterpretation: "...indicating that the proteolytic activity of ADAM17 is indeed necessary...". TAPI-1 containing food will surely inhibit more proteases than just ADAM-17.

This has now been amended.

5. page 13. The two citations of Fig 4G in the Discussion appear to be out of context and do not to make sense. Please clarify.

We thank the reviewer for pointing this out. We have now removed the incorrect citation.

6. The Figure 8 model refers to neurons and glia, not photoreceptors and PGCs. This expansion of the paradigm from the eye to the generic nervous system is not justified from the data presented.

This has now been amended.

NON-ESSENTIAL:

1. Strengthening the ADAM17 - TNF connection in the *in vivo* context of retinal degeneration would be valuable, since this is the major new finding to the existing PGC-photoreceptor degeneration models. A key prediction is that PGC expression (at or near endogenous levels) of cleaved Eiger or activated Grindelwald will rescue the ADAM17^{-/-} phenotype.

This is indeed an interesting area for future exploration, although we feel that it is beyond the scope of this piece of work. The reviewer will of course be aware of the work from Pierre Leopold's group that shows that ADAM17 does indeed release a soluble form of Eiger, which, in that case, acts as a long-range adipokine.

Reviewer 2:

In this study, Mulyil and colleagues report that the TNF signaling cascade in Drosophila, mediated by ADAM17, Eiger and Grnd, protects pigment glial cells (PGC)s of the eye from degeneration. They specifically show that ADAM17 mutants exhibit a progressive retinal degeneration phenotype, and show that this reflects ADAM17's requirement in the PGCs. The authors report a similar phenotype with loss of function mutants of Eiger (Drosophila TNFalpha family ligand) and Grnd (TNF receptor). Evidence presented here indicates that in ADAM17 mutants, the PGCs accumulate Reactive Oxygen Species (ROS) in the mitochondria, which causes the activation of JNK signaling to induce lipid biogenesis for lipid droplet (LD) formation. Breaking down LDs early on suppresses the degeneration phenotype. However, the onset of retinal degeneration correlates with the dispersal of LD droplets, indicating that there are two temporal phases of LD formation that affects PGC degeneration in distinct ways. The authors use iPSC cells to demonstrate that ADAM17 has similar roles in mammalian cultured cells.

Overall, this is a very nicely executed study that establishes a previously unrecognized role of ADAM17 in the protection of cells from ROS and LD-mediated neurodegeneration. The quality of the data is very high, with proper controls and validation of critical observations through multiple independent means. I only have a few minor suggestions that could further enhance the clarity of the manuscript.

1. *The authors used two guide RNAs (gRNA) to generate an ADAM17 mutant allele. The methods section describing this mutant is rather brief, mentioning that the mutants were screened using a "melting curve analysis." The authors should explain whether the mutant line that they used has an actual deletion between the two gRNA targeted sites or not. Also, it would be useful to give the mutant allele a name.*

We have now added the relevant details to the Methods section.

2. *The authors do not describe the source of Eiger, Grnd, and wgn loss-of-function mutant alleles used in Figure 4 (also Figure S6) in the Methods section. Which alleles did they use?*

We thank the reviewer for pointing the oversight. We have now added the relevant details to the Methods section.

3. In Figure 4L - P, the authors show the effect of *Eiger*, *grnd*, and *wgn* loss on neuronal degeneration at two weeks of age. Why was this time point chosen when their analyses of *adam17* loss-of-function were primarily done at five weeks of age (e.g., Figures 1 and 3)? The degeneration of *Eiger* mutants at two weeks of age, as shown in Figure 4, is somewhat modest. I wonder if the authors could report a more clear difference if they were to examine five-week old flies.

In fact we did include the five week degeneration data for *Eiger*, *Grindelwald* and *wengen* mutants. They are shown in Figure EV4N-P.

Dear Matthew,

Thank you for submitting your revised manuscript to The EMBO Journal.

Your study has now been seen by referee #1 and the comments are provided below. The referee appreciates the introduced changes and supports publication here. There is just one remaining discussion point that should be resolved.

When you submit your revised manuscript will you please also address the following points.

- You have at the moment 8 keywords but can only have 5.
- I think there are figure callouts missing to Fig 5Q,R. Please also correct callouts for the appendix figures and add "Appendix" to S figures
- We don't allow data not shown please see page 6 and 38.
- We require a data availability section. I don't think any data needs to be deposited in an external data base and please add this study includes no data deposited in external repositories
- We include a synopsis of the paper (see <http://emboj.embopress.org/>). Please provide me with a general summary statement and 3-5 bullet points that capture the key findings of the paper.
- We also need a summary figure for the synopsis. The size should be 550 wide by 400 high (pixels). You can also use something from the figures if that is easier.
- I have asked our publisher to do their pre-publication checks on the paper. They will send me the file within the next few days. Please wait to upload the revised version until you have received their comments.

That should be all - let me know if you have any further questions.

Best Karin

Karin Dumstrei, PhD
Senior Editor
The EMBO Journal

- a point-by-point response to the referees' comments, with a detailed description of the changes made (as a word file).

- a word file of the manuscript text.

- individual production quality figure files (one file per figure)

- a complete author checklist, which you can download from our author guidelines

(<https://www.embopress.org/page/journal/14602075/authorguide>).

- Expanded View files (replacing Supplementary Information)

Further information is available in our Guide For Authors:

The revision must be submitted online within 90 days; please click on the link below to submit the revision online before 27th Aug 2020.

Referee #1:

The authors have now fully addressed all of the major and minor criticisms of the original manuscript, except for major point 1 and minor point 2. Both points relate to the central critical issue of whether the relevant source of excess lipids is glia or neurons

Major point 1: The authors response states "All our data make the point strongly that ADAM17 is required only in glia, not photoreceptor neurons, to protect against retinal degeneration. Reviewer 1 does not question this. The more specific question is the source of the lipids that contribute the glial LD accumulation: might they be made in adjacent neurons and transported to the glia? Our best evidence against this is that expression of the lipase Brummer in glia prevents the accumulation of LDs and neurodegeneration; in contrast, expression of lipase in neurons had no measurable effect on LD accumulation. This implies that the excess lipids are not substantially being generated by neurons. Significantly, it contrasts with the results for lipase overexpression in similar experiments by the by the Bellen group, again highlighting the differences between the two contexts. Nevertheless, when we looked at the less direct measure of retinal degeneration, we did see a low level of rescue when Brummer was expressed in neurons. Since we saw no effect on LD accumulation itself, it is hard to interpret this as a clear sign of neuronal involvement as a source of accumulating lipids, but we acknowledge that we cannot rule out a minor contribution."

I agree that all the evidence points to a PGC-specific role for ADAM17 in this system. However, the evidence for a PGC-specific origin of the excess lipids is far less clear. Brummer is one of many factors that could regulate lipid release from photoreceptors, so the absence of a LD phenotype in the PR>Bmm experiment is very weak evidence in favour of a PGC-specific lipid origin. Since the interesting protective role of ADAM17/TNF does not depend on the origin of the excess lipids, why do the authors not leave the question open? The current coronavirus situation does indeed preclude further experimental tests (e.g I had suggested lipid carrier/receptor experiments). Given this limitation, I request that the discussion point "...implying that the primary source of accumulating lipids is the glial cells." be changed to a less definitive conclusion.

Minor point 2: The authors response states "We believe that majority of the ROS is neuronally generated. This is because in the absence of light, which is expected to lower specifically neuronal ROS, the degeneration in ADAM17 mutants is substantially reduced. But we have amended the text of the abstract to make the claim less sweeping."

The belief of the authors seems strange as the experiment using glial overexpression of SOD2 fully rescued LD accumulation and neurodegeneration (better than with neuronal overexpression of SOD2). Anyhow, the text amendments here seem to suffice as they do leave open the possible contributions from both neurons and glia.

EMBOJ-2020-104415. Muliyl et al. Responses to final review comments

Major Point 1

As requested, we have altered our discussion about the source of the lipids that generate the excess lipid droplets in the glial cells. The conclusion is now less definitive about our favoured model.

Minor Point 2

Although the reviewer discusses our response, s/he finishes by saying that our earlier amendments suffice.

Dear Matthew,

Thank you for submitting your revised manuscript to The EMBO Journal. I have now had a chance to take a look at it and appreciate the introduced changes.

I am therefore very pleased to accept the manuscript for publication here. Congratulations on a very nice study

with best wishes

Karin

Karin Dumstrei, PhD
Senior Editor
The EMBO Journal

Please note that it is EMBO Journal policy for the transcript of the editorial process (containing referee reports and your response letter) to be published as an online supplement to each paper. If you do NOT want this, you will need to inform the Editorial Office via email immediately. More information is available here: http://emboj.embopress.org/about#Transparent_Process

Your manuscript will be processed for publication in the journal by EMBO Press. Manuscripts in the PDF and electronic editions of The EMBO Journal will be copy edited, and you will be provided with page proofs prior to publication. Please note that supplementary information is not included in the proofs.

Should you be planning a Press Release on your article, please get in contact with embojournal@wiley.com as early as possible, in order to coordinate publication and release dates.

If you have any questions, please do not hesitate to call or email the Editorial Office. Thank you for your contribution to The EMBO Journal.

** Click here to be directed to your login page: <http://emboj.msubmit.net>

Corresponding Author Name: Matthew Freeman

Journal Submitted to: EMBO J

Manuscript Number: EMBOJ-2020-104415